# Fractal Landscapes in Policy Optimization

**Tao Wang**
UC San Diego
taw003@ucsd.edu

**Sylvia Herbert**
UC San Diego
sherbert@ucsd.edu

**Sicun Gao**
UC San Diego
sicung@ucsd.edu

## Abstract

Policy gradient lies at the core of deep reinforcement learning (RL) in continuous domains. Despite much success, it is often observed in practice that RL training with policy gradient can fail for many reasons, even on standard control problems with known solutions. We propose a framework for understanding one inherent limitation of the policy gradient approach: the optimization landscape in the policy space can be extremely non-smooth or fractal for certain classes of MDPs, such that there does not exist gradient to be estimated in the first place. We draw on techniques from chaos theory and non-smooth analysis, and analyze the maximal Lyapunov exponents and Hölder exponents of the policy optimization objectives. Moreover, we develop a practical method that can estimate the local smoothness of objective function from samples to identify when the training process has encountered fractal landscapes. We show experiments to illustrate how some failure cases of policy optimization can be explained by such fractal landscapes.

## 1 Introduction

Deep reinforcement learning has achieved much success in various applications [23, 30, 38], but they also often fail, especially in continuous spaces, on control problems that other methods can readily solve. The understanding of such failure cases is still limited. For instance, the training process of reinforcement learning is unstable and the learning curve can fluctuate during training in ways that are hard to predict. The probability of obtaining satisfactory policies can also be inherently low in reward-sparse or highly nonlinear control tasks. Existing analysis of the failures focuses on limitations of the sampling and optimization algorithms, such as function approximation errors [35, 39], difficulty in data collection [34], and aggressive updates in the policy space [28]. There has not been much study of potentially deeper causes of failures that may be inherent in the formulation of policy optimization problems.

Motivated by the common observation that small updates in the policy parameters can significantly change the performance, we analyze the smoothness of the optimization landscapes in policy optimization. Drawing on chaos theory, we introduce the concept of maximal Lyapunov exponent (MLE) [17] to the RL setting to measure the exponential rate of trajectory divergence in MDP. It seems contradictory that a trajectory in chaotic systems can be both exponentially divergent and uniformly bounded at the same time, and we will show that these two conflicting facts combine to yield the fractal structure in the optimization landscape. Intuitively, the objective function is non-differentiable when the rate of trajectory divergence exceeds the decay rate of discount factor. Furthermore, this finding indicates that the fluctuations observed in the loss curve are not just due to the numerical or sampling error but rather reflect the intrinsic properties of the corresponding MDP.

We should emphasize that the fractal landscapes that we will demonstrate are stronger than various existing results on the non-smoothness [2, 7]. Most non-smooth objectives that have been studied still assume is local Lipschitz continuity or piecewise smoothness that implies differentiability *almost everywhere* (such as $f(x) = |x|$). Instead, by showing that the loss landscape of policy optimization can be fractal, we demonstrate the absence of descent directions, which causes the failure of first-order

37th Conference on Neural Information Processing Systems (NeurIPS 2023).

methods in general. Since such behavior is an intrinsic property of the underlying dynamical systems, the results show fundamental limitations of policy gradient methods on certain classes of MDPs.

The paper is organized as follows. In Section 3 and 4, we will introduce the preliminaries and develop the theory for deterministic policies. In particular, we show that the optimization landscape is fractal, even when all elements within the MDP are deterministic. Next, we consider stochastic policies and provide an example to show how non-smoothness can still occur if without additional assumptions. In Section 5, we turn the theoretical analysis into a practical sampling-based method for estimating the Hölder exponent to determine whether the optimization objective is differentiable at a specific parameter vector. It can also indicate if the training process has encountered fractal regions by comparing the regression slope with some fixed threshold. In Section 6, we show experiments that demonstrate and compare the landscapes of different MDPs.

## 2 Related work

**Policy gradient and Q-learning methods.** Policy gradient methods [33, 41] formulate RL as an optimization problem in the parameter space, with many variations such as natural policy gradient [16], deterministic policy gradient [29], deep deterministic policy gradient [18], trust region policy optimization [27] and proximal policy optimization [28], were proposed. As all of these algorithms aim to estimate the gradient of the objective function over the policy parameters, they become ill-posed when the objective is non-differentiable, which is the focus of our analysis.

Another popular approach for model-free RL is Q-learning methods, which approximate the Q-function of the policy at each step [22, 40]. As neural networks become more and more popular, they are employed as function approximators in deep Q-learning algorithms [9, 13, 37]. Since the foundation of Q-learning methods is established upon the estimation of value functions, a poor approximation can completely ruin the entire training process. In this paper, we will show that the value functions in a certain class of MDPs exhibit significant non-smoothness, making them challenging to represent using existing methods.

**Chaos in machine learning.** Chaotic behaviors due to randomness in the learning dynamics have been reported in other learning problems [6, 21, 25]. For instance, when training recurrent neural networks for a long period, the outcome behaves like a random walk due to the problems of vanishing and the exploding gradients [4]. It served as motivation for the work [24], which points out that the chaotic behavior in finite-horizon model-based reinforcement learning problems may be caused by long chains of nonlinear computation. A similar observation was made in [31]. However, we show that in RL, the objective function is provably smooth if the time horizon is finite and the underlying dynamics is differentiable. Instead, we focus on the general context of infinite-horizon problems in MDPs, in which case the objective function can become non-differentiable.

**Loss landscape of policy optimization.** It has been shown that the objective functions in finite state-space MDPs are smooth [1, 42], which enables the use of gradient-based methods and direct policy search. It also explains why the classical RL algorithms in [32] are provably efficient in finite space settings. Also, such smoothness results can be extended to some continuous state-space MDPs with special structures. For instance, the objective function in Linear Quadratic Regulator (LQR) problems is almost smooth [10] as long as the cost is finite. Similar results are obtained for the $\mathcal{H}_2/\mathcal{H}_\infty$ problem [43]. For the robust control problem, although the objective function may not be smooth, it is locally Lipschitz continuous, which implies differentiability *almost everywhere*, and further leads to global convergence of direct policy search [11]. There is still limited theoretical study of loss landscapes of policy optimization for nonlinear and complex MDPs. We aim to partially address this gap by pointing out the possibility that the loss landscape can be highly non-smooth and even fractal, which is far more complex than the previous cases.

## 3 Preliminaries

### 3.1 Dynamical Systems as Markov Decision Processes

We consider Markov Decision Processes (MDPs) that encode continuous control problems for dynamical systems defined by difference equations of the form:

$$s_{t+1} = f(s_t, a_t), \tag{1}$$

where $s_t \in \mathcal{S} \subset \mathbb{R}^n$ is the state at time $t$, $s_0$ is the initial state and $a_t \sim \pi_\theta(\cdot|s_t) \in \mathcal{A} \subset \mathbb{R}^m$ is the action taken at time $t$ based on a policy parameterized by $\theta \in \mathbb{R}^p$. We assume that both the state space $\mathcal{S}$ and the action space $\mathcal{A}$ are compact. The objective function of the RL problem to minimize is defined by $V^{\pi_\theta}$ of policy $\pi_\theta$:

$$J(\theta) = V^{\pi_\theta}(s_0) = \mathbb{E}_{a_t \sim \pi_\theta(\cdot|s_t)}[\sum_{t=0}^{\infty} \gamma^t c(s_t, a_t)], \tag{2}$$

where $\gamma \in (0,1)$ is the discount factor and $c(s,a)$ is the cost function. The following assumptions are made throughout this paper:

- (A.1) $f : \mathbb{R}^n \times \mathbb{R}^m \to \mathbb{R}^n$ is Lipschitz continuous over any compact domains (i.e., locally Lipschitz continuous);

- (A.2) The cost function $c : \mathbb{R}^n \times \mathbb{R}^m \to \mathbb{R}$ is non-negative and locally Lipschitz continuous everywhere;

- (A.3) The state space is closed under transitions, i.e., for any $(s,a) \in \mathcal{S} \times \mathcal{A}$, the next state $s' = f(s,a) \in \mathcal{S}$.

### 3.2 Policy gradient methods

Policy gradient methods estimate the gradient of the objective $J(\cdot)$ with respect to the parameters of the policies. A commonly used form is

$$\nabla J(\theta) = \mathbb{E}_{a_t \sim \pi_\theta(\cdot|s_t)}[\nabla_\theta \log \pi_\theta(a_t|s_t) \, A^{\pi_\theta}(s_t, a_t)], \tag{3}$$

where $\pi_\theta(\cdot|\cdot)$ is a stochastic policy parameterized by $\theta$. $A^{\pi_\theta}(s,a) = Q^{\pi_\theta}(s,a) - V^{\pi_\theta}(s)$ is the advantage function often used for variance reduction and $Q^{\pi_\theta}(\cdot, \cdot)$ is the $Q$-value function of $\pi_\theta$. The theoretical guarantee of the convergence of policy gradient methods is typically established by the argument that the tail term $\gamma^t \nabla_\theta V^{\pi_\theta}(s)$ diminishes as $t$ increases, for any $s \in \mathcal{S}$ [33]. For such claims to hold, two assumptions are needed:

- $\nabla_\theta V^{\pi_\theta}(s)$ exists and is continuous for all $s \in \mathcal{S}$;
- $\|\nabla_\theta V^{\pi_\theta}(s)\|$ is uniformly bounded over $\mathcal{S}$.

The second assumption is automatically satisfied if the first assumption holds in the case that $\mathcal{S}$ is either finite or compact. However, as we will see in Section 4 and 6, the existence of $\nabla_\theta V^{\pi_\theta}(\cdot)$ may fail in many continuous MDPs even if $\mathcal{S}$ is compact, which challenges the fundamental well-posedness of policy gradient methods.

### 3.3 Maximal Lyapunov Exponents

Behaviors of chaotic systems have sensitive dependence on their initial conditions. To be precise, consider the system $s_{t+1} = F(s_t)$ with initial state $s_0 \in \mathbb{R}^n$, and suppose that a small perturbation $\Delta Z_0$ is made to $s_0$. The divergence from the original trajectory of the system under this perturbation at time $t$, say $\Delta Z(t)$, can be estimated by $\|\Delta Z(t)\| \simeq e^{\lambda t}\|\Delta Z_0\|$ with some $\lambda$ that is called the Lyapunov exponent. For chaotic systems, Lyapunov exponents are typically positive, which implies an exponential divergence rate of the separation of nearby trajectories [19]. Since the Lyapunov exponent at a given point may depend on the direction of the perturbation $\Delta Z_0$, and we are interested in identifying the largest divergence rate, the maximal Lyapunov exponent (MLE) is formally defined as follows:

**Definition 3.1.** *(Maximal Lyapunov exponent) For the dynamical system $s_{t+1} = F(s_t)$, $s_0 \in \mathbb{R}^n$, the maximal Lyapunov exponent $\lambda_{\max}$ at $s_0$ is defined as the largest value such that*

$$\lambda_{\max} = \limsup_{t \to \infty} \limsup_{\|\Delta Z_0\| \to 0} \frac{1}{t} \log \frac{\|\Delta Z(t)\|}{\|\Delta Z_0\|}. \tag{4}$$

Note that systems with unstable equilibria, not necessarily chaotic, can have positive MLEs.

### 3.4 Fractal Landscapes

The Hausdorff dimension is the most fundamental concept in fractal theory. We first introduce the concept of $\delta$-cover and Hausdorff measure:

**Definition 3.2.** *($\delta$-cover) Let $\{U_i\}$ be a countable collection of sets of diameter at most $\delta$ (i.e. $|U_i| = \sup\{\|x - y\| : x, y \in U_i\} \leq \delta$) and $F \subset \mathbb{R}^N$, then $\{U_i\}$ is a $\delta$-cover of $F$ if $F \subset \cup_{i=1}^{\infty} U_i$.*

**Definition 3.3.** *(Hausdorff measure) For any $F \subset \mathbb{R}^N$ and $s \geq 0$, let*

$$\mathcal{H}_{\delta}^s(F) = \inf\{\sum_{i=1}^{\infty} |U_i|^s : \{U_i\} \text{ is a } \delta\text{-cover of } F\}.$$

*Then we call the limit $\mathcal{H}^s(F) = \lim_{\delta \to 0} \mathcal{H}_{\delta}^s(F)$ the s-dimensional Hausdorff measure of F.*

The definition of Hausdorff dimension follows immediately:

**Definition 3.4.** *(Hausdorff dimension) Let $F \subset \mathbb{R}^N$ be a subset, then its Hausdorff dimension*

$$\dim_H F = \inf\{s \geq 0 : \mathcal{H}^s(F) = 0\} = \sup\{s \geq 0 : \mathcal{H}^s(F) = \infty\}.$$

And we introduce the notion of $\alpha$-Hölder continuity that extends the concept of Lipschitz continuity:

**Definition 3.5.** *($\alpha$-Hölder continuity) Let $\alpha > 0$ be a scalar. A function $g : \mathbb{R}^N \to \mathbb{R}$ is $\alpha$-Hölder continuous at $x \in \mathbb{R}^N$ if there exist $C > 0$ and $\delta > 0$ such that*

$$|g(x) - g(y)| \leq C\|x - y\|^{\alpha}$$

*for all $y \in \mathcal{B}(x, \delta)$, where $\mathcal{B}(x, \delta)$ denotes the open ball of radius $\delta$ centered at $x$.*

The definition reduces to Lipschitz continuity when $\alpha = 1$. A function is not differentiable, if the largest Hölder exponent at a given point is less than 1. Just as smoothness is commonly associated with Lipschitz continuity, fractal behavior is closely related to Hölder continuity. In particular, for an open set $F \subset \mathbb{R}^k$ and a continuous mapping $\eta : F \to \mathbb{R}^p$ with $p > k$, the image set $\eta(F)$ is fractal when its Hausdorff dimension $\dim_H \eta(F)$ is strictly greater than $k$, which occurs when $\eta : F \to \mathbb{R}^p$ is $\alpha$-Hölder continuous with exponent $\alpha < 1$:

**Proposition 3.1.** *([8]) Let $F \subset \mathbb{R}^k$ be a subset and suppose that $\eta : F \to \mathbb{R}^p$ is $\alpha$-Hölder continuous where $\alpha > 0$, then $\dim_H \eta(F) \leq \frac{1}{\alpha} \dim_H F$.*

It implies that if the objective function is $\alpha$-Hölder for some $\alpha < 1$, its loss landscape $\mathcal{L}_J = \{(\theta, J(\theta)) \in \mathbb{R}^{N+1} : \theta \in \mathbb{R}^N\}$ can be fractal. Further discussion of the theory on fractals can be found in Appendix C.

## 4 Fractal Landscapes in the Policy Space

In this section, we will show that the objective $J(\theta)$ in policy optimization can be non-differentiable when the system has positive MLEs. We will first consider Hölder continuity of $V^{\pi_\theta}(\cdot)$ and $J(\cdot)$ with deterministic policies in 4.1 and 4.2, and then discuss the case of stochastic policies in 4.3.

### 4.1 Hölder Exponent of $V^{\pi_\theta}(\cdot)$

We first consider a deterministic policy $\pi_\theta$ that maps states to actions $a = \pi_\theta(s)$ instead of distributions. Consider a fixed policy parameter $\theta \in \mathbb{R}^p$ such that the MLE of (1), namely $\lambda(\theta)$, is greater than $-\log \gamma$. Let $s_0' \in \mathcal{S}$ be another initial state that is close to $s_0$, i.e., $\delta = \|s_0' - s_0\| > 0$ is small enough. According to the assumption (A.3) and the compactness of the state space, we can find a constant $M > 0$ such that both $\|s_t\| \leq M$ and $\|s_t'\| \leq M$ for all $t \in \mathbb{N}$, where $\{s_t\}_{t=1}^{\infty}$ and $\{s_t'\}_{t=1}^{\infty}$ are the trajectories starting from $s_0$ and $s_0'$, respectively. Motivated by (4), we further make the following assumptions:

- (A.4) There exists $K_1 > 0$ such that $\|s_t' - s_t\| \leq K_1 \delta e^{\lambda(\theta)t}$ for all $t \in \mathbb{N}$ and $\delta = \|s_0' - s_0\| > 0$.
- (A.5) The policy $\pi : \mathbb{R}^N \times \mathbb{R}^n \to \mathbb{R}^m$ is locally Lipschitz continuous everywhere.

We then have following theorem, and it provides a lower bound for the Hölder exponent of $J$ whose detailed proof can be found in Appendix B.1.

**Theorem 4.1.** *(Non-smoothness of $V^{\pi_\theta}$) Assume (A.1)-(A.5) and the parameterized policy $\pi_\theta(\cdot)$ is deterministic. Let $\lambda(\theta)$ denote the MLE of* (1) *at $\theta \in \mathbb{R}^N$. Suppose that $\lambda(\theta) > -\log \gamma$, then $V^{\pi_\theta}(\cdot)$ is $\frac{-\log \gamma}{\lambda(\theta)}$-Hölder continuous at $s_0$.*

**Proof sketch of Theorem 4.1:**  Suppose that $p \in (0, 1]$ is some constant for which we would like to prove that $V^{\pi_\theta}(s)$ is $p$-Hölder continuous at $s = s_0$, and here we take $p = -\frac{\log \gamma}{\lambda(\theta)}$.

According to Definition 3.5, it suffices to find some $C' > 0$ such that $|V^{\pi_\theta}(s_0') - V^{\pi_\theta}(s_0)| \leq C' \delta^p$ when $\delta = \|s_0 - s_0'\| \ll 1$. Consider the relaxed form

$$|V^{\pi_\theta}(s_0') - V^{\pi_\theta}(s_0)| \leq \sum_{t=0}^{\infty} \gamma^t |c(s_t, \pi_\theta(s_t)) - c(s_t', \pi_\theta(s_t'))| \leq C' \delta^p. \tag{5}$$

Now we split the entire series into three parts as shown in Figure 1: the sum of first $T_2$ terms, the sum from $t = T_2 + 1$ to $T_3 - 1$, and the sum from $t = T_3$ to $\infty$. First, applying (A.4) to the sum of the first $T_2$ terms yields

$$\sum_{t=0}^{T_2} \gamma^t |c(s_t, \pi_\theta(s_t)) - c(s_t', \pi_\theta(s_t'))| \leq \frac{e^{(\lambda(\theta)+\log \gamma)T_2}}{1-\gamma} K_1 K_2 \delta, \tag{6}$$

where $K_2 > 0$ is the Lipschitz constant obtained by (A.2) and (A.5). If we wish to bound the right-hand side of (6) by some term of order $\mathcal{O}(\delta^p)$ when $\delta \ll 1$, the length $T_2(\delta) \in \mathbb{N}$ should satisfy

$$T_2(\delta) \simeq C_1 + \frac{p-1}{\lambda(\theta) + \log \gamma} \log(\delta), \tag{7}$$

where $C_1 > 0$ is some constant independent of $p$ and $\delta$.

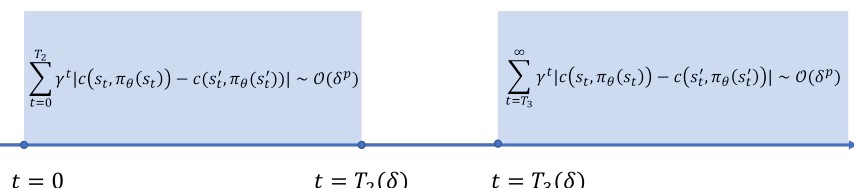

Figure 1: An illustration of the two series (7) and (9) that need to cover the entire $\mathbb{R}$ when $\delta \to 0$.

Next, for the sum of the tail terms in $V^{\pi_\theta}(\cdot)$ starting from $T_3 \in \mathbb{N}$, it is automatically bounded by

$$\sum_{t=T_3}^{\infty} \gamma^t |c(s_t, \pi_\theta(s_t)) - c(s_t', \pi_\theta(s_t'))| \leq \frac{2M_2 e^{T_3 \log \gamma}}{1-\gamma}, \tag{8}$$

where $M_2 = \max_{s \in \mathcal{S}} c(s, \pi_\theta(s))$ is the maximum of continuous function $c(\cdot, \pi_\theta(\cdot))$ over the compact domain $\mathcal{S}$ (and hence exists). if we bound the right-hand side of (8) by a term of order $\mathcal{O}(\delta^p)$, it yields

$$T_3(\delta) \simeq C_2 + \frac{p}{\log \gamma} \log(\delta), \tag{9}$$

for some independent constant $C_2 > 0$. Since the sum of (6) and (8) provides a good estimate of $V^{\pi_\theta}$ only if $T_3(\delta) - T_2(\delta) \leq N_0$ for some $N_0 > 0$ as $\delta \to 0$, otherwise there would be infinitely many terms in the middle as $\delta \to 0$ that cannot be controlled by any $\mathcal{O}(\delta^p)$ terms. In this case, we have

$$(C_2 - C_3) + \left(\frac{p}{\log \gamma} - \frac{p-1}{\lambda(\theta) + \log \gamma}\right) \log(\delta) \leq N_0, \tag{10}$$

as $\log(\delta) \to -\infty$, which implies that the slopes satisfy the inequality

$$\frac{p}{\log \gamma} - \frac{p-1}{\lambda(\theta) + \log \gamma} \geq 0, \tag{11}$$

where the equality holds when $p = -\frac{\log \gamma}{\lambda(\theta)}$. Thus, $V^{\pi_\theta}(s)$ is $-\frac{\log \gamma}{\lambda(\theta)}$-Hölder continuous at $s = s_0$.

On the other hand, the following counterexample shows that Theorem 4.1 has provided the strongest Hölder-continuity result for $V^{\pi_\theta}(s)$ at $s = s_0$ under the assumptions (A.1)-(A.5):

**Example 4.1.** *Consider a one-dimensional MDP $s_{t+1} = f(s_t, a_t)$ where*

$$f(s, a) = \begin{cases} -1, & a \leq -1, \\ a, & -1 < a < 1, \\ 1, & a \geq 1, \end{cases} \tag{12}$$

*with state space $\mathcal{S} = [-1, 1]$ and cost function $c(s, a) = |s|$. Let the policy be linear, namely $\pi_\theta(s) = \theta \cdot s$ and $\theta \in \mathbb{R}$. It can be verified that all assumptions (A.1)-(A.5) are satisfied. Now let $s_0 = 0$ and $\theta > 1$, then applying (4) directly yields*

$$\lambda(\theta) = \limsup_{t \to \infty} \limsup_{\|\Delta Z_0\| \to 0} \frac{1}{t} \log \frac{\|\Delta Z(t)\|}{\|\Delta Z_0\|} = \limsup_{t \to \infty} \limsup_{\|\Delta Z_0\| \to 0} \frac{1}{t} \log \frac{\|\Delta Z_0\| \theta^t}{\|\Delta Z_0\|} = \log \theta.$$

*Let $\delta > 0$ be sufficiently small, then*

$$V^{\pi_\theta}(\delta) = \sum_{t=0}^{\infty} \delta \gamma^t \theta^t = \sum_{t=0}^{T_0(\delta)} \delta \gamma^t \theta^t + \sum_{t=T_0(\delta)}^{\infty} \gamma^t \geq \frac{\gamma^{T_0(\delta)}}{1 - \gamma}$$

*where $T_0(\delta) = 1 + \lfloor \frac{-\log \delta}{\log \theta} \rfloor \in \mathbb{N}$ and $\lfloor \cdot \rfloor$ is the flooring function. Therefore, we have*

$$|V^{\pi_\theta}(\delta) - V^{\pi_\theta}(0)| = V^{\pi_\theta}(\delta) \geq \frac{\gamma^{\frac{-\log \delta}{\log \theta} + 1}}{1 - \gamma} = \frac{\gamma}{1 - \gamma} \delta^{\frac{-\log \gamma}{\log \theta}}.$$

**Remark 4.1.** *Another way to see why it is theoretically impossible to prove $p$-Hölder continuity for $V^{\pi_\theta}$ for any $p > -\frac{\log \gamma}{\lambda(\theta)}$: notice that the inequality (10) no longer holds as $\log \delta \to -\infty$ since*

$$\frac{p}{\log \gamma} - \frac{p - 1}{\lambda(\theta) + \log \gamma} < 0.$$

*Thus, $p = \frac{-\log \gamma}{\lambda(\theta)}$ is the largest Hölder exponent of $V^{\pi_\theta}(\cdot)$ that can be proved in the worst case.*

**Remark 4.2.** *The value function $V^{\pi_\theta}(s)$ is Lipschitz continuous at $s = s_0$ when the maximal Lyapunov exponent $\lambda(\theta) < -\log \gamma$, since there exists a constant $K'$ such that*

$$\begin{aligned} |V^{\pi_\theta}(s_0') - V^{\pi_\theta}(s_0)| &\leq \sum_{t=0}^{\infty} \gamma^t |c(s_t, \pi_\theta(s_t)) - c(s_t', \pi_\theta(s_t'))| \\ &\leq \sum_{t=0}^{\infty} \gamma^t K' \delta e^{\lambda(\theta)t} \\ &\leq \delta K' \sum_{t=0}^{\infty} e^{(\lambda(\theta) + \log \gamma)t} \\ &\leq \frac{K' \delta}{1 - e^{(\lambda(\theta) + \log \gamma)}} \end{aligned}$$

*where $\delta = \|s_0 - s_0'\| > 0$ is the difference in the initial state.*

### 4.2 Hölder Exponent of $J(\cdot)$

The following lemma establishes a direct connection between $J(\theta)$ and $J(\theta')$ through value functions:

**Lemma 4.1.** *Suppose that $\theta, \theta' \in \mathbb{R}^p$, then*

$$V^{\pi_{\theta'}}(s_0) - V^{\pi_\theta}(s_0) = \sum_{t=0}^{\infty} \gamma^t (Q^{\pi_\theta}(s_t^{\theta'}, \pi_{\theta'}(s_t^{\theta'})) - V^{\pi_\theta}(s_t^{\theta'}))$$

*where $\{s_t^{\theta'}\}_{t=0}^{\infty}$ is the trajectory generated by the policy $\pi_{\theta'}(\cdot)$.*

The proof can be found in the Appendix B.2. Notice that indeed we have $J(\theta') = V^{\pi_{\theta'}}(s_0)$ and $J(\theta) = V^{\pi_\theta}(s_0)$, substituting with these two terms in the previous lemma and doing some calculations lead to the following main theorem whose proof can be found in the Appendix B.3:

**Theorem 4.2.** *(Non-smoothness of $J$) Assume (A.1)-(A.5) and the parameterized policy $\pi_\theta(\cdot)$ is deterministic. Let $\lambda(\theta)$ denote the MLE of (1) at $\theta \in \mathbb{R}^p$. Suppose that $\lambda(\theta) > -\log\gamma$, then $J(\cdot)$ is $\frac{-\log\gamma}{\lambda(\theta)}$-Hölder continuous at $\theta$.*

**Remark 4.3.** *In fact, the set of assumptions (A.1)-(A.5) is quite general and does not exclude the case of constant cost functions $c(s, a) \equiv const$, which always results in a smooth landscape regardless of the underlying dynamics, even though they are rarely used in practice. However, recall that the $\frac{-\log\gamma}{\lambda(\theta)}$-Hölder continuity is a result of exponential divergence of nearby trajectories, when a cost function can continuously distinguish two separate trajectories (e.g., quadratic costs) with a discount factor close to 1, the landscape will be fractal as shown in Section 6. Another way to see it is to look into the relaxation in (5) where the Hölder continuity is obtained from the local Lipschitz continuity of $c(s, a)$, i.e., $|c(s, \pi_\theta(s)) - c(s', \pi_\theta(s'))| \le K_2\|s - s'\|$. Therefore, the Hölder continuity is tight if for any $\delta > 0$, there exists $s_0' \in \mathcal{B}(s_0, \delta)$ such that $|c(s_t, \pi_\theta(s_t)) - c(s_t', \pi_\theta(s_t'))| \ge K_3\|s_t - s_t'\|$ with some $K_3 > 0$ for all $t \in \mathbb{N}$. We will leave the further investigation for future studies.*

The following example illustrates how the smoothness of loss landscape changes with $\lambda(\theta)$ and $\gamma$:

**Example 4.2.** *(Logistic model) Consider the following MDP:*

$$s_{t+1} = (1 - s_t)a_t, \quad s_0 = 0.9, \tag{13}$$

*where the policy $a_t$ is given by deterministic linear function $a_t = \pi_\theta(s_t) = \theta s_t$. The objective function is defined as $J(\theta) = \sum_{t=0}^{\infty} \gamma^t (s_t^2 + 0.1 a_t^2)$ where $\gamma \in (0, 1)$ is the discount factor. It is well-known that (13) begins to exhibit chaotic behavior with positive MLEs (as shown in Figure 2a) when $\theta \ge 3.3$ [15], so we plot the graphs of $J(\theta)$ for different discount factors over the interval $\theta \in [3.3, 3.9]$. From Figure 2b to 2d, the non-smoothness becomes more and more significant as $\gamma$ grows. In particular, Figure 2e shows that the value of $J(\theta)$ fluctuates violently even within a very small interval of $\theta$, suggesting a high degree of non-differentiability in this region.*

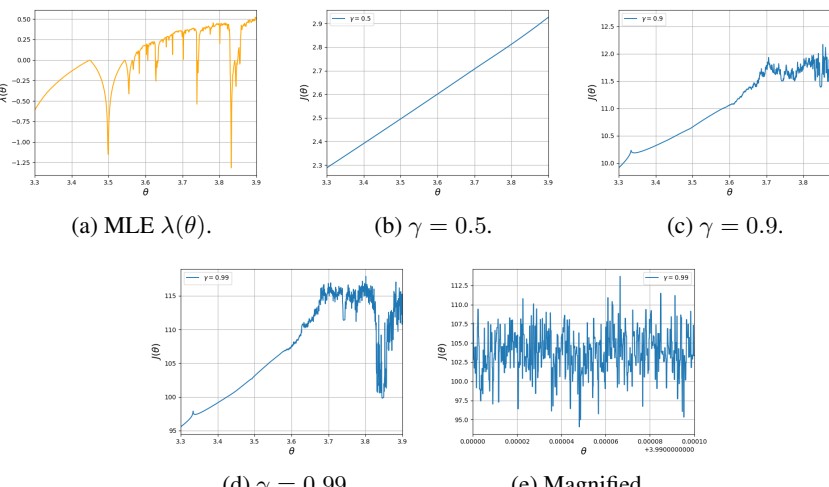

Figure 2: The value of MLE $\lambda(\theta)$ for $\theta \in [3.3, 3.9]$ is shown in 2a. The graph of objective function $J(\theta)$ for different values of $\gamma$ are shown in 2b-2e where $J(\theta)$ is estimated by the sum of first 1000 terms in the infinite series.

## 4.3 Stochastic Policies

The original MDP (1) becomes stochastic when a stochastic policy is employed. First, let us consider the slightly modified version of MLE for stochastic policies:

$$\tilde{\lambda}_{max} = \limsup_{t \to \infty} \limsup_{\|\Delta Z_0\| \to 0} \frac{1}{t} \log \frac{\|\mathbb{E}_\pi[\Delta Z_\omega(t)]\|}{\|\Delta Z_0\|}. \tag{14}$$

where $\Delta Z_0 = s_0' - s_0$ is a small pertubation made to the initial state and $\Delta Z_\omega(t) = s_t'(\omega) - s_t(\omega)$ is the difference in the sample path at time $t \in \mathbb{N}$ and sample $\omega \in \Omega$. Since this definition is consistent with that in (4) when sending the variance to 0, we use the same notation $\lambda(\theta)$ to denote the MLE at given $\theta \in \mathbb{R}^N$ and again assume $\lambda(\theta) > -\log \gamma$. Since policies in most control and robotics environments are deterministic, this encourages the variance to converge to 0 during training.

However, unlike the deterministic case where the Hölder continuity result was proved under the assumption that the policy $\pi_\theta(s)$ is locally Lipschitz continuous, stochastic policies instead provide a probability distribution from which the action is sampled. Thus, a stochastic policy cannot be locally Lipschitz continuous in $\theta$ when approaching its deterministic limit. For instance, consider the one-dimensional Gaussian distribution $\pi_\theta(a|s)$ where $\theta = [\mu, \sigma]^T$ denotes the mean and variance. As the variance $\sigma$ approaches 0, $\pi_\theta(a|s)$ becomes more and more concentrated at $a = \mu s$, and eventually converges to the Dirac delta function $\delta(a - \mu s)$, which means that $\pi_\theta(a|s)$ cannot be Lipschitz continuous within a neighborhood of any $\theta = [\mu, 0]^T$ even though its deterministic limit $\pi_\theta(s) = \mu s$ is indeed Lipschitz continuous. The following example illustrates that in this case, the Hölder exponent of the objective function $J(\cdot)$ can still be less than 1:

**Example 4.3.** *Suppose that the one-dimensional MDP $s_{t+1} = f(s_t, a_t)$ where $f(s, a)$ is defined as in (12) over the state space $\mathcal{S} = [-1, 1]$ and action space $\mathcal{A} = [0, \infty)$. The cost function is $c(s, a) = s + 1$. Also, the parameter space is $\theta = [\theta_1, \theta_2]^T \in \mathbb{R}^2$ and the policy $\pi_\theta(\cdot|s) \sim \mathcal{U}(|\theta_1|s + |\theta_2|, |\theta_1|s + 2|\theta_2|)$ is a uniform distribution. It is easy to verify that all required assumptions are satisfied. Let the initial state $s_0 = 0$ and $\theta_1 > 1, \theta_2 = 0$, then applying (14) directly yields $\lambda(\theta) = \log \theta_1$ similarly as in Example 4.1. Now suppose that $\theta_2' > 0$ is small and $\theta' = [\theta_1, \theta_2']^T$, then for any $\omega \in \Omega$ in the sample space, the sampled trajectory $\{s_t'\}$ generated by $\pi_{\theta'}$ has*

$$s_{t+1}'(\omega) \geq \theta_1 s_t'(\omega) + \theta_2' > \theta_1 s_t'(\omega) \geq \theta_1^t s_1'(\omega) \geq \theta_1^t(\theta_2')$$

*when $s_{t+1}'(\omega) < 1$. Thus, we have $s_{t+1}'(\omega) = 1$ for all $\omega \in \Omega$ and $t \geq T_0(\theta') = 1 + \lfloor \frac{-\log \theta_2'}{\log \theta_1} \rfloor$, which further leads to*

$$J(\theta') = \frac{1}{1-\gamma} + \sum_{t=0}^{\infty} \gamma^t \, \mathbb{E}_{\pi_{\theta'}}[s_t'] \geq J(\theta) + \sum_{t=T_0(\delta)}^{\infty} \gamma^t \, \mathbb{E}_{\pi_{\theta'}}[s_t'] \geq \frac{\gamma}{1-\gamma}(\theta_2')^{\frac{-\log \gamma}{\log \theta_1}}$$

*using the fact that $J(\theta) = \frac{1}{1-\gamma}$. Plugging $\|\theta - \theta'\| = \theta_2'$ into the above inequality yields*

$$J(\theta') - J(\theta) \geq \frac{\gamma}{1-\gamma}\|\theta' - \theta\|^{\frac{-\log \gamma}{\log \theta_1}}. \tag{15}$$

*where the Hölder exponent is again $\frac{-\log \gamma}{\lambda(\theta)}$ as in Example 4.1.*

**Remark 4.4.** *Consider the 1-Wasserstein distance as defined in [36] between the distribution $\delta(a - \mu s)$ and $\mathcal{U}(|\theta_1|s + |\theta_2|, |\theta_1|s + 2|\theta_2|)$, which is given by $W_1(\theta_1, \theta_2) = \frac{3|\theta_2|}{2}$. It is Lipschitz continuous at $\theta_2 = 0$, even though the non-smooth result in (15) holds. Therefore, probability distribution metrics, such as the Wasserstein distance, are too "coarse" to capture the full fractal nature of the objective function. This also suggests that further assumptions regarding the pointwise smoothness of probability density functions are necessary to create a smooth landscape with stochastic policies, even though they may exclude the case of $\sigma \to 0$ as discussed earlier.*

## 5 Estimating Hölder Exponents from Samples

In the previous sections, we have seen that the objective function $J(\theta)$ can be highly non-smooth and thus gradient-based methods may not work well in the policy parameter space. The question is: how can we determine whether the objective function $J(\theta)$ is differentiable at some $\theta = \theta_0$ or not in high-dimensional settings? Note that $J(\theta)$ may have different levels of smoothness along different directions. To address it, we propose a statistical method to estimate the Hölder exponent. Consider the objective function $J(\theta)$ and a probability distribution whose variance is finite. Consider the isotropic Gaussian distribution $X \sim \mathcal{N}(\theta_0, \sigma^2 \mathcal{I}_p)$ where $\mathcal{I}_p$ is the $p \times p$ identity matrix. For continuous objective function $J(\cdot)$, then its variance matrix can be expressed as

$$Var(J(X)) = \mathbb{E}_{X \sim \mathcal{N}(\theta_0, \sigma^2 \mathcal{I})}[J(X) - \mathbb{E}_{X \sim \mathcal{N}(\theta_0, \sigma^2 \mathcal{I}_p)}[J(X)]]^2]$$
$$= \mathbb{E}_{X \sim \mathcal{N}(\theta_0, \sigma^2 \mathcal{I}_p)}[(J(X) - J(\xi'))^2]$$

where $\xi' \in \mathbb{R}^p$ is obtained from applying the intermediate value theorem to $\mathbb{E}_{X \sim \mathcal{N}(\theta_0, \sigma^2 \mathcal{I}_p)}[J(X)]$ and hence not a random variable. If $J(\cdot)$ is locally Lipschitz continuous at $\theta_0$, say $|J(\theta) - J(\theta_0)| \leq K\|\theta - \theta_0\|$ for some $K > 0$ when $\|\theta - \theta_0\|$ is small, then it has the following approximation

$$Var(J(X)) \leq K^2 \mathbb{E}_{X \sim \mathcal{N}(\theta_0, \sigma^2 \mathcal{I}_p)}[\|X - \xi'\|^2] \simeq (Var(X))^2 \sim \mathcal{O}(\sigma^2) \tag{16}$$

when $\sigma \ll 1$. Therefore, (16) provides a way to directly determine whether the Hölder exponent of $J(\cdot)$ at any given $\theta \in \mathbb{R}^p$ is less than 1, especially when the dimension $p$ is large. In particular, taking the logarithm on both sides of (16) yields

$$\log Var_\sigma(J(X)) \leq C + 2\log\sigma \tag{17}$$

for some constant $C$ where the subscript in $Var_\sigma(J(X))$ indicates its dependence on the standard deviation $\sigma$ of $X$. Thus, the log-log plot of $Var_\sigma(J(X))$ versus $\sigma$ is expected to be close to a straight line with slope $k \geq 2$ when $J(\theta)$ is locally Lipschitz continuous around $\theta = \theta_0$. Therefore, one can determine the smoothness by sampling around $\theta_0$ with different variances and estimating the slope via linear regression. Usually, $J(\theta)$ is Lipschitz continuous at $\theta = \theta_0$ when the slope $k$ is close to or greater than 2, and it is non-differentiable if the slope is less than 2.

## 6  Experiments

In this section, we will validate the theory presented in this paper through common RL tasks. All environments are adopted from The OpenAI Gym Documentation [5] with continuous control input. The experiments are conducted in two steps: first, we randomly sample a parameter $\theta_0$ from a Gaussian distribution and estimate the gradient $\eta(\theta_0)$ from (3); second, we evaluate $J(\theta)$ at $\theta = \theta_0 + \delta\eta(\theta_0)$ for each small $\delta > 0$. According to our results, the loss curve is expected to become smoother as $\gamma$ decreases, since smaller $\gamma$ makes the Hölder exponent $\frac{-\log\gamma}{\lambda(\theta)}$ larger. In the meantime, the policy gradient method (3) should give a better descent direction while the true objective function $J(\cdot)$ becoming smoother.

Notice that a single sample path can always be non-smooth when the policy is stochastic and hence interferes the desired observation, we use stochastic policies to estimate the gradient in (3), and apply their deterministic version (by setting variance equal to 0) when evaluating $J(\theta)$. Regarding the infinite series, we use the sum of first 1000 terms to approximate $J(\theta)$. The stochastic policy is given by $\pi_\theta(\cdot|s) \sim \mathcal{N}(u(s), \sigma^2 \mathcal{I}_p)$ where the mean $u(s)$ is represented by the 2-layer neural network $u(s) = W_2 \tanh(W_1 s)$ where $W_1 \in \mathcal{M}_{r \times n}(\mathbb{R})$ and $W_2 \in \mathcal{M}_{m \times r}(\mathbb{R})$ are weight matrices. Let $\theta = [W_1, W_2]^T$ denote the vectorized policy parameter. For the width of the hidden layer, we use $r = 8$ for the inverted pendulum and acrobot, and $r = 64$ for the hopper.

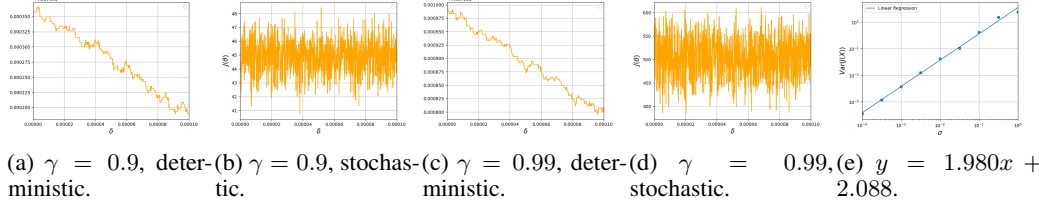

(a) $\gamma = 0.9$, deterministic.  (b) $\gamma = 0.9$, stochastic.  (c) $\gamma = 0.99$, deterministic.  (d) $\gamma = 0.99$, stochastic.  (e) $y = 1.980x + 2.088$.

Figure 3: The experimental results of inverted pendulum. In 3e, the linear regression result is obtained for $\gamma = 0.9$. The loss curves $J(\theta)$ are presented in 3a-3d where $\theta = \theta_0 + \delta\eta(\theta_0)$ with step size $10^{-7}$.

**Inverted Pendulum.**   The inverted pendulum task is a standard test case for RL algorithms, and here we use it as an example of non-chaotic system. The initial state is always taken as $s_0 = [-1, 0]^T$ ($[0, 0]^T$ is the upright position), and quadratic cost function $c(s, a) = s_t^T Q s_t + 0.001\|a_t\|^2$, where $Q = \text{diag}(1, 0.1)$ is a $2 \times 2$ diagonal matrix, $s_t \in \mathbb{R}^2$ and $a_t \in \mathbb{R}$. The initial parameter is given by $\theta_0 \sim \mathcal{N}(0, 0.05^2 \mathcal{I})$. In Figure 4a and 4c, we see that the loss curve is close to a straight line within a very small interval, which indicates the local smoothness of $\theta_0$. It is validated by the estimate of the Hölder exponent of $J(\theta)$ at $\theta = \theta_0$ which is based on (16) by sampling many parameters around $\theta_0$ with different variance. In Figure 3e, the slope $k = 1.980$ is very closed to 2 so Lipschitz continuity (and hence differentiability) is verified at $\theta = \theta_0$. As a comparison, the loss curve of single random sample path is totally non-smooth as shown in Figure 3b and 3d.

**Acrobot.** The acrobot system is well-known for its chaotic behavior and hence we use it as the main test case. Here we use the cost function $c(s, a) = s_t^T Q s_t + 0.005\|a_t\|^2$, where $Q = \text{diag}(1, 1, 0.1, 0.1)$, $s_t \in \mathbb{R}^4$ and $a_t \in \mathbb{R}$. The initial state is $s_0 = [1, 0, 0, 0]^T$. The initial parameter is again sampled from $\theta_0 \sim \mathcal{N}(0, 0.05^2 \mathcal{I})$. From Figure 4a-4c, the non-smoothness grows as $\gamma$ increases and finally becomes completely non-differentiable when $\gamma = 0.99$ which is the most common value used for discount factor. It partially explains why the acrobot task is difficult to policy gradient methods. In Figure 4e, the Hölder exponent of $J(\theta)$ at $\theta = \theta_0$ is estimated as $\alpha \simeq 0.43155 < 1$, which further indicates non-differentiability around $\theta_0$.

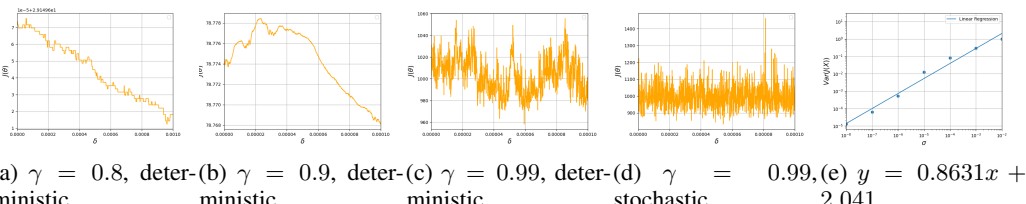

(a) $\gamma = 0.8$, deter-ministic.  (b) $\gamma = 0.9$, deter-ministic.  (c) $\gamma = 0.99$, deter-ministic.  (d) $\gamma = 0.99$, stochastic.  (e) $y = 0.8631x + 2.041$

Figure 4: The experimental results of acrobot. In Figure 4e, the linear regression result is obtained for $\gamma = 0.9$. The loss curves $J(\theta)$ are presented in 4a-4d where $\theta = \theta_0 + \delta\eta(\theta_0)$ with step size $10^{-7}$.

**Hopper.** Now we consider the Hopper task in which the cost function is defined $c(s, a) = (1.25 - s[0]) + 0.001\|a\|^2$, where $s[0]$ is the first coordinate in $s \in \mathbb{R}^{11}$ which indicates the height of hopper. Because the number of parameters involved in the neural network is larger, the initial parameter is instead sampled from $\theta_0 \sim \mathcal{N}(0, 10^2 \mathcal{I})$. As we see that in Figure 5a, the loss curve is almost a straight line when $\gamma = 0.8$, and it starts to exhibit non-smoothness when $\gamma = 0.9$ and becomes totally non-differentiable when $\gamma = 0.99$. A supporting evidence by the Hölder exponent estimation is provided in Figure 5e where the slope is far less than 2.

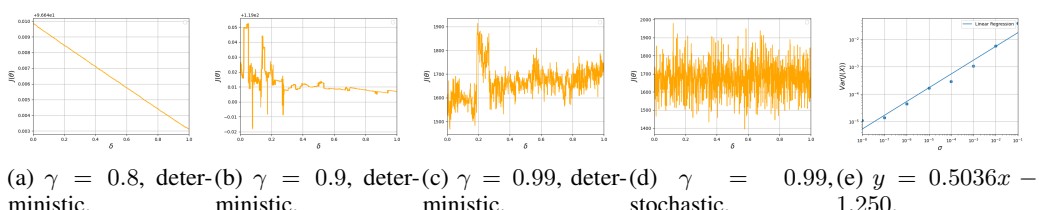

(a) $\gamma = 0.8$, deter-ministic.  (b) $\gamma = 0.9$, deter-ministic.  (c) $\gamma = 0.99$, deter-ministic.  (d) $\gamma = 0.99$, stochastic.  (e) $y = 0.5036x - 1.250$.

Figure 5: The experimental results of hopper. In Figure 5e, the linear regression result is obtained for $\gamma = 0.9$. The loss curves $J(\theta)$ are presented in 5a-5d where $\theta = \theta_0 + \delta\eta(\theta_0)$ with step size $10^{-3}$.

# 7   Conclusion

In this paper, we initiate the study of chaotic behavior in reinforcement learning, especially focusing on how it is reflected on the fractal landscape of objective functions. A method to statistically estimate the Hölder exponent at some given parameter is proposed, so that one can figure out if the training process has encountered fractal landscapes or not. We believe that the theory established in this paper can help to explain many existing results in reinforcement learning, such as the hardness of complex control tasks and the fluctuating behavior of training curves. It also poses a serious question to the well-posedness of policy gradient methods given the fact that no gradient exists in many continuous state-space RL problems. Being aware of the fact that the non-smoothness of loss landscapes is an intrinsic property of the model, rather than a consequence of any numerical or statistical errors, we conjecture that the framework developed in this paper might provide new insights into the limitations of a wider range of deep learning problems beyond the realm of reinforcement learning.

## 8 Acknowledgements

Our work is supported by NSF Career CCF 2047034, NSF AI Institute CCF 2112665, ONR YIP N00014-22-1-2292, NSF CCF DASS 2217723, and Amazon Research Award. The authors thank Zichen He, Bochao Kong and Xie Wu for insightful discussions.

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

# A  A Brief introduction to chaos theory

As mentioned in the Introduction, chaos exists in many systems in the real world. Although no universal definition of chaos can be made, there are, indeed, three features that a chaotic system usually possesses [14]:

- *Dense periodic points;*

- *Topological transitivity;*

- *Sensitive dependence on initial conditions;*

In some cases, some of these properties imply the others. It is important to note that, despite the appearance of chaos is always accompanied by high unpredictability, *the chaotic behavior is entirely deterministic* and is not a consequence of randomness. Another interesting fact is that trajectories in a chaotic system are usually bounded, which drives us to think about the convergence of policy gradient methods beyond the boundedness of state spaces.

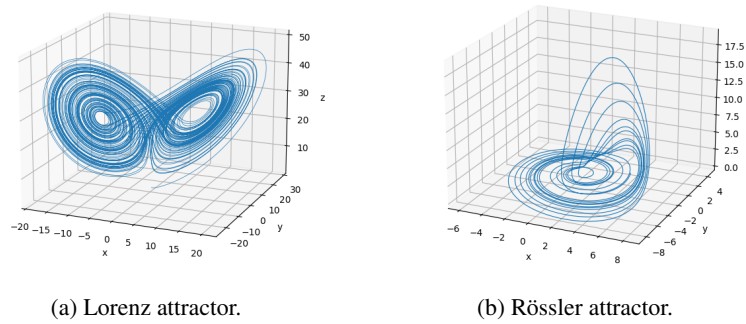

(a) Lorenz attractor.                                      (b) Rössler attractor.

Figure 6: The Lorenz system and Rössler system are standard examples of chaotic systems, in which a small perturbation in the initial state can result in a significant divergence in the entire trajectory.

Actually, it can be summarized from the results in this paper that for a given MDP, the following three features contribute most to its chaotic behavior:

- *Infinite time horizon ($t \to \infty$);*

- *Continuous state space ($\|\Delta Z_0\| \to 0$);*

- *Exponential divergence ($\lambda_{max} > 0$);*

Since these features are not necessarily bound to certain types of continuous state-space MDPs, it would be exciting for future studies to investigate other types of MDPs using the framework developed in the paper.

# B  Proofs omitted in Section 4

## B.1  Proof of Theorem 4.1

*Proof.* Suppose that $s_0' \in \mathcal{S}$ is another initial state close to $s_0$ and $\delta = \|s_0 - s_0'\|$. Let $T_1 \in \mathbb{N}$ be the smallest integer that satisfies

$$T_1 \geq \frac{1}{\lambda(\theta)} \log(\frac{2M_2}{K_1 \delta}), \tag{18}$$

where $M_2 = 1 + \max_{s \in \mathcal{S}} c(s, \pi_\theta(s)) > 0$ is the maximum of the continuous function $c(\cdot, \pi_\theta(\cdot))$ over $\mathcal{S}$, then applying the Lipschitz condition of $c(\cdot, \pi_\theta(\cdot))$ yields

$$
\begin{aligned}
\sum_{t=0}^{T_1} \gamma^t |c(s_t, \pi_\theta(s_t)) - c(s'_t, \pi_\theta(s'_t))| &\leq \sum_{t=0}^{T_1} \gamma^t K_2 \|s_t - s'_t\| \\
&\leq \sum_{t=0}^{T_1} K_1 K_2 e^{(\lambda(\theta) + \log \gamma)t} \delta \\
&\leq K_1 K_2 \delta \frac{e^{\frac{\lambda(\theta) + \log \gamma}{\lambda(\theta)} \log(\frac{2M_2}{K_1 \delta}) + 2(\lambda(\theta) + \log \gamma)}}{e^{(\lambda(\theta) + \log \gamma)} - 1} \\
&= \frac{e^{2(\lambda(\theta) + \log \gamma)} K_2 K_1^{\frac{-\log \gamma}{\lambda(\theta)}} (2M_2)^{1 + \frac{\log \gamma}{\lambda(\theta)}}}{e^{(\lambda(\theta) + \log \gamma)} - 1} \delta^{\frac{-\log \gamma}{\lambda(\theta)}}
\end{aligned}
$$

where $K_2 > 0$ is the Lipschitz constant of $c(\cdot, \pi_\theta(\cdot))$ over compact set $\mathcal{S}$.

On the other hand, the tail terms in $J(\cdot)$ is bounded by

$$
\begin{aligned}
\sum_{t=T_1+1}^{\infty} \gamma^t |c(s_t, \pi_\theta(s_t)) - c(s'_t, \pi_\theta(s'_t))| &\leq \sum_{t=T_1+1}^{\infty} 2M_2 \gamma^t \\
&\leq \sum_{t=T_1}^{\infty} 2M_2 \gamma^t \\
&= 2M_2 \frac{\gamma^{T_1}}{1 - \gamma} \\
&\leq \frac{2M_2}{1 - \gamma} \left(\frac{K_1}{2M_2}\right)^{\frac{-\log \gamma}{\lambda(\theta)}} \delta^{\frac{-\log \gamma}{\lambda(\theta)}}
\end{aligned}
$$

using $|c(s_t, \pi_\theta(s_t)) - c(s'_t, \pi_\theta(s'_t))| \leq 2M_2$. Combining the above two inequalities yields

$$
\begin{aligned}
&|V^{\pi_\theta}(s'_0) - V^{\pi_\theta}(s_0)| \\
&\leq \sum_{t=0}^{\infty} \gamma^t |c(s_t, \pi_\theta(s_t)) - c(s'_t, \pi_\theta(s'_t))| \\
&= \sum_{t=0}^{T_1} \gamma^t |c(s_t, \pi_\theta(s_t)) - c(s'_t, \pi_\theta(s'_t))| + \sum_{t=T_1+1}^{\infty} \gamma^t |c(s_t, \pi_\theta(s_t)) - c(s'_t, \pi_\theta(s'_t))| \\
&\leq \left(\frac{e^{2(\lambda(\theta) + \log \gamma)} K_2 K_1^{\frac{-\log \gamma}{\lambda(\theta)}} (2M_2)^{1 + \frac{\log \gamma}{\lambda(\theta)}}}{e^{(\lambda(\theta) + \log \gamma)} - 1} + \frac{2M_2}{1 - \gamma} \left(\frac{K_1}{2M_2}\right)^{\frac{-\log \gamma}{\lambda(\theta)}}\right) \delta^{\frac{-\log \gamma}{\lambda(\theta)}}
\end{aligned}
$$

and we complete the proof. $\qquad\square$

## B.2 Proof of Lemma 4.1

*Proof.* For the ease of notation, let $s_t = s_t^\theta$, $s'_t = s_t^{\theta'}$, $u(s) = \pi_\theta(s)$ and $u'(s) = \pi_{\theta'}(s)$.

$$V^{\pi_{\theta'}}(s_0) - V^{\pi_\theta}(s_0) = \sum_{t=0}^{\infty} \gamma^t c(s'_t; u'(s'_t)) - V^{\pi_\theta}(s_0)$$

$$= \sum_{t=0}^{\infty} \gamma^t (c(s'_t; u'(s'_t)) + V^{\pi_\theta}(s'_t) - V^{\pi_\theta}(s'_t)) - V^{\pi_\theta}(s_0)$$

$$= \sum_{t=0}^{\infty} \gamma^t (c(s'_t; u'(s'_t)) + \gamma V^{\pi_\theta}(s'_{t+1}) - V^{\pi_\theta}(s'_t) + V^{\pi_\theta}(s'_t) - \gamma V^{\pi_\theta}(s'_{t+1})) - V^{\pi_\theta}(s_0)$$

$$= \sum_{t=0}^{\infty} \gamma^t (Q^{\pi_\theta}(s'_t, u'(s'_t)) - V^{\pi_\theta}(s'_t)) + \sum_{t=0}^{\infty} \gamma^t (V^{\pi_\theta}(s'_t) - \gamma V^{\pi_\theta}(s'_{t+1})) - V^{\pi_\theta}(s_0).$$

Using the fact that $\gamma^t V^{\pi_\theta}(x'_{t+1}) \to 0$ as $t \to \infty$ from (A.3) yields

$$V^{\pi_{\theta'}}(s_0) - V^{\pi_\theta}(s_0) = \sum_{t=0}^{\infty} \gamma^t (Q^{\pi_\theta}(s'_t, u'(s'_t)) - V^{\pi_\theta}(s'_t)) + V^{\pi_\theta}(s'_0) - V^{\pi_\theta}(s_0)$$

$$= \sum_{t=0}^{\infty} \gamma^t (Q^{\pi_\theta}(s'_t, u'(s'_t)) - V^{\pi_\theta}(s'_t))$$

and the proof is completed using $s'_0 = s_0$. $\qquad \square$

### B.3 Proof of Theorem 4.2

*Proof.* First, we will show that $Q^{\pi_\theta}(s, a)$ is $\frac{-\log \gamma}{\lambda(\theta)}$-Hölder continuous with respect to $a$. Note that for any given $a \in \mathcal{A}$ and any $a' \in \mathcal{A}$ such that $\|a - a'\| \ll 1$,

$$|Q^{\pi_\theta}(s, a) - Q^{\pi_\theta}(s, a')| \leq |c(s, a) - c(s, a')| + \gamma |V^{\pi_\theta}(f(s, a)) - V^{\pi_\theta}(f(s, a'))|$$

$$\leq K_1 \|a - a'\| + \gamma \|f(s, a) - f(s, a')\|^{\frac{-\log \gamma}{\lambda(\theta)}}$$

$$\leq K_1 \|a - a'\| + \gamma K_2 \|a - a'\|^{\frac{-\log \gamma}{\lambda(\theta)}}$$

$$\leq K_3 \|a - a'\|^{\frac{-\log \gamma}{\lambda(\theta)}}$$

for some $K_3 > 0$ using the locally Lipschitz continuity of $c$ and $f$.

Note that $V^{\pi_\theta}(s) = Q^{\pi_\theta}(s, \pi_\theta(s))$, combining it with Lemma 4.1 yields

$$|J(\theta') - J(\theta)| \leq \sum_{t=0}^{\infty} \gamma^t |Q^{\pi_\theta}(s'_t, u'(s'_t)) - V^{\pi_\theta}(s'_t)|$$

$$= \sum_{t=0}^{\infty} \gamma^t |Q^{\pi_\theta}(s'_t, \pi_{\theta'}(s'_t)) - Q^{\pi_\theta}(s'_t, \pi_\theta(s'_t))|$$

$$\leq \sum_{t=0}^{\infty} \gamma^t K_3 \|\pi_{\theta'}(s'_t) - \pi_\theta(s'_t)\|^{\frac{-\log \gamma}{\lambda(\theta)}}$$

$$\leq \sum_{t=0}^{\infty} \gamma^t K_3 K_4 \|\theta' - \theta\|^{\frac{-\log \gamma}{\lambda(\theta)}}$$

$$= \frac{K_3 K_4}{1 - \gamma} \|\theta' - \theta\|^{\frac{-\log \gamma}{\lambda(\theta)}}$$

using the fact that $\pi_\theta(s)$ is Lipschitz continuous in a neighborhood of $(\theta, s) \in \mathbb{R}^N \times \mathcal{S}$ for some constant $K_4 > 0$ and we complete the proof. $\qquad \square$

# C  From the perspective of fractal theory

We will go through some basic concepts in fractal theory that are related to the study of non-smooth functions.

## C.1  The Hausdorff dimension

We will show that the Hausdorff dimension is well-defined: First, it is clear that when $\delta < 1$, $\mathcal{H}^s_\delta(F)$ is non-increasing with respect to $s$. Thus, $\mathcal{H}^s(F)$ is non-increasing as well. Let $s \geq 0$ such that $\mathcal{H}^s(F) < \infty$, then for any $t > s$ and any $\delta$-cover $\{U_i\}$ of $F$, we have

$$\sum_{i=1}^\infty |U_i|^t \leq \delta^{t-s} \sum_{i=1}^\infty |U_i|^s$$

which implies $\mathcal{H}^t(F) = 0$ by taking infimum on both sides and letting $\delta \to 0$. Therefore, the set $\{s \geq 0 : 0 < \mathcal{H}^s(F) < \infty\}$ contains at most one point, which further implies $\inf\{s \geq 0 : \mathcal{H}^s(F) = 0\} = \sup\{s \geq 0 : \mathcal{H}^s(F) = \infty\}$.

More details regarding the well-posedness of Hausdorff dimension can be found in [3, 8]. In particular, one can easily verify that the Hausdorff dimension coincides with the standard dimension (i.e. $s \in \mathbb{N}$) when $F$ is a regular manifold. Typically, the Hausdorff dimension of a fractal is not an integer, and we will be exploiting this fact through the section. A famous example is the Weierstrass function as shown in Figure 7. A comparison of Figure 2e and Figure 7c (they have the same scale) gives some sense about how non-smooth the objective function can be in practice.

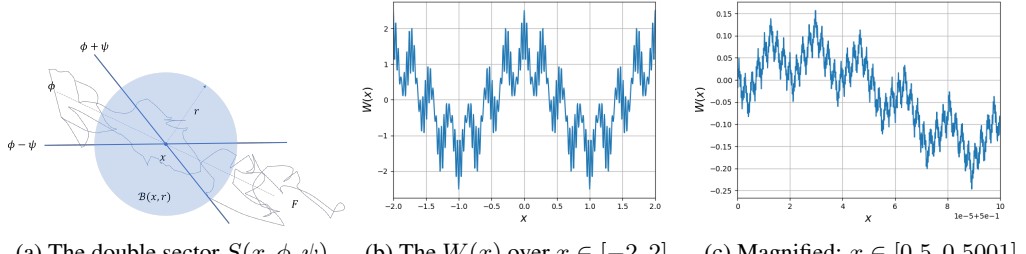

(a) The double sector $S(x, \phi, \psi)$.  (b) The $W(x)$ over $x \in [-2, 2]$.  (c) Magnified: $x \in [0.5, 0.5001]$.

Figure 7: (a) shows how the double sector looks like. In (b) and (c), the Weierstrass function is given by $W(x) = \sum_{n=0}^\infty a^n \cos(b^n \pi x)$ where $a = 0.6, b = 7$. The Hausdorff dimension of its loss curve is calculated as $\dim_H \mathcal{L}_W = 2 + \log_b a \simeq 1.73$. Also, according to [12], such $W(x)$ is nowhere differentiable when $0 < a < 1$ and $ab \geq 1$.

## C.2  Non-existence of tangent plane

Actually, when $J(\cdot)$ is Lipschitz continuous on any compact subset of $\mathbb{R}^N$, by the Rademacher's Theorem, we know that it is differentiable almost everywhere which implies the existence of tangent plane. As it comes to fractal landscapes, however, the tangent plane itself does not exist for almost every $\theta \in \mathbb{R}^N$, which makes all policy gradient algorithms ill-posed. Although similar results were obtained for higher-dimensional cases as in [26], we focus on the two-dimensional case so that it provides a more direct geometric intuition. First, we introduce the notion of $s$-sets:

**Definition C.1.** *Let $F \subset \mathbb{R}^2$ be a Borel set and $s \geq 0$, then $F$ is called an s-set if $0 < \mathcal{H}^s(F) < \infty$.*

The intuition is that: when the dimension of fractal $F$ is a fraction between 1 and 2, then there is no direction along which a significant part of $F$ concentrates within a small double sector with vertex $x$ as shown in Figure 7a. To be precise, let $S(x, \phi, \psi)$ denote the double sector and $r > 0$, then we say that $F$ has a tangent at $x \in F$ if there exists a direction $\phi$ such that for every angle $\phi > 0$, it has

1. $\limsup_{r \to 0} \frac{\mathcal{H}^s(F \cap \mathcal{B}(x,r))}{(2r)^s} > 0$;

2. $\lim_{r \to 0} \frac{\mathcal{H}^s(F \cap (\mathcal{B}(x,r) \backslash S(x,\phi,\psi)))}{(2r)^s} = 0$;

where the first condition states that the set $F$ behaves like a fractal around $x$, and the second condition implies that the part of $F$ lies outside of any double sector $S(x, \phi, \psi)$ is negligible when $r \to 0$. Then, the main result is as follows:

**Proposition C.1.** *(Non-existence of tangent planes, [8]) If $F \subset \mathbb{R}^2$ is an s-set with $1 < s < 2$, then at almost all points of $F$, no tangent exists.*

Therefore, "estimate the gradient" no longer makes sense since there does not exist a tangent line/plane at almost every point on the loss surface. This means that all policy gradient algorithms are ill-posed since there is no gradient for them to estimate at all.

### C.3 Accumulated uncertainty

Another issue that may emerge during training process is the accumulation of uncertainty. To see how the uncertainty entered at each step accumulates and eventually blows up when generating a path along fractal boundaries, let us consider the following toy problem: Suppose that the distance between the initial point $\theta_0 \in \mathbb{R}^N$ and the target $\theta^*$ is $d > 0$, and step size $\delta_k > 0$ is adapted at the $k$-th step, as shown in Figure 8a. If there exists $c > 0$ such that the projection $\langle \theta^* - \theta_0, \theta_{k+1} - \theta_k \rangle \geq cd\delta_k$ for all $k \in \mathbb{N}$ which implies that the angle between the direction from $\theta_k$ to $\theta_{k+1}$ and the true direction $\theta^* - \theta_0$ does not exceed $\arccos(c)$, in this case, a successful path $\{\theta_k\}$ that converges to $\theta^*$ should give

$$\sum_{k=0}^{\infty} cd\delta_k \leq \sum_{k=0}^{\infty} \langle \theta^* - \theta_0, \theta_{k+1} - \theta_k \rangle = \langle \theta^* - \theta_0, \theta^* - \theta_0 \rangle = d^2$$

using $\theta_k \to \theta^*$ as $k \to \infty$, which is equivalent to $\sum_{k=0}^{\infty} \delta_k \leq \frac{d}{c}$.

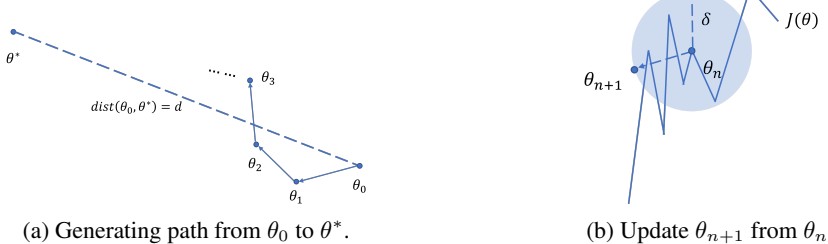

(a) Generating path from $\theta_0$ to $\theta^*$.      (b) Update $\theta_{n+1}$ from $\theta_n$.

Figure 8: Illustrations of the statistical challenges in implementing policy gradient algorithms on a fractal loss surface.

On the other hand, when walking on the loss surface, it is not guaranteed to follow the correct direction precisely all the time. For any small step size $\delta > 0$, the uncertainty fraction $u(\delta)$ involved in every single step can be estimated by the following result [20]:

**Proposition C.2.** *Let $\delta > 0$ be the step size and $\beta = N + 1 - \dim_H J$ where $\dim_H J$ is the Hausdorff dimension of loss surface of $J(\cdot)$, then the uncertainty $u(\delta) \sim \mathcal{O}(\delta^\beta)$ when $\delta \ll 1$.*

Therefore, we may assume that there exists another $c' > 0$ such that the uncertainty $U_k$ at the $k$-th step has $U_k \leq c'\delta_k^\beta$ for all $k = 0, 1, \dots$. Then, the accumulated uncertainty

$$U = \sum_{k=0}^{\infty} U_k \leq c' \sum_{k=0}^{\infty} \delta_k^\beta$$

is bounded when $\beta = 1$ (i.e. boundary is smooth) using the earlier result $\sum_{k=0}^{\infty} \delta_k \leq \frac{d}{c}$. However, the convergence of $\sum_{k=0}^{\infty} \delta_k$ no longer guarantees the convergence of $\sum_{k=0}^{\infty} \delta_k^\beta$ when $\beta < 1$, and a counterexample is the following series:

$$\delta_k = \frac{1}{k(\log(k+2))^2}$$

for all $k = 0, 1, \dots$, which implies the uncertainty accumulated over the course of iterations may increase dramatically and eventually cause the sequence $\theta_k$ to become random when walking on fractal boundaries.

