# OpenReview forum: "Fractal Landscapes in Policy Optimization"
_NeurIPS.cc/2023/Conference — NeurIPS 2023 poster_

### Official Review · Reviewer_Qd1b · 2023-06-19

**Soundness:** 2 fair
**Presentation:** 2 fair
**Contribution:** 3 good
**Rating:** 4
**Confidence:** 3

**Summary:**

This paper studies the dynamics of policy gradient training in
reinforcement learning, and argues that the policy gradient often does
not exist due to the lack of regularity of the training objective. The
authors derive a threshold on the "maximum Lypanunov exponent", a
quantity they introduce which captures the sensitivity of trajectories
to policy parameters, beyond which sufficient regularity (and existence
of the policy gradient) cannot be guaranteed. Beyond this, the authors
demonstrate how the Holder coefficient of the training objective as a
function of the policy parameters can be estimated from sample
trajectories, and validate via numerical simulations that the Holder
coefficient estimated from samples correlates with the stability of
policy gradient training.


**Strengths:**

This work studies an interesting an important problem that deep RL is
faced with. The chaos-theory-inspired approach taken by the authors is
relevant, and in particular, it is nice to see how the Holder exponent
can be approximated from samples. The experimental results are fairly
interesting, demonstrating a clear correlation between the estimated
Holder coefficient and the stability of training.

**Weaknesses:**

As I understand it, the main theoretical result of the paper is that for
a small enough Lyapunov exponent, we can prove that the policy gradient
objective is smooth enough, but not the converse. That is, for a large
enough Maximum Lyapunov Exponent, it is shown that sometimes the
objective is non-smooth, but clearly it may also still be smooth
(consider for example any MDP with a constant reward function). It is
claimed on line 197 that when the Maximum Lyapunov Exponent is above the
threshold, the objective is "very unlikely" to be Lipschitz continuous,
but I do not find any convincing justification for this beyond
intuition.

Moreover, even if the objective is not Lipschitz continuous, it is not
clear to me that this should necessarily raise alarms with regard to
policy optimization. A function that is non-diffentiable at even one
point (that is, a set of measure zero!) is not Lipschitz continuous, but
I would not expect this to necessarily pose any severe optimization
problems for policy gradient algorithms. If the objective turns out to
be fractal (as suggested by the title), that would probably be much more
severe, but fractals are not really discussed at all in the paper. The
authors at one point claim that a lower Holder exponent *can* imply a
"more fractal" landscape, but no evidence is given that loss landscapes
tend to be fractal in the RL problems that we encounter.

Furthermore, while I find the method of estimating the Holder exponent
from samples interesting, there are many assumptions made (such as the
"tightness" of the Holder continuity of the objective) that I would not
necessarily expect to hold in practice. Consequently, it is not clear to
me that the constant extracted from this scheme would be an accurate
representation of the Holder exponent.

The Maximum Lyapunov Exponent introduced in this work is abbreviated to
MLE, which should probably be avoided.


**Questions:**

Line 57 says "Consequently, it is not surprising that [DDPG, TRPO,
PPO, etc] often have a low probability of obtaining a satisfactory
solution". Is this true? Some of these algorithms have proven to be
quite successful (albeit very sensitive).

In section 3.3, the "divergence" terms $\Delta Z(t)$ should be defined
explicitly. Moreover, the definition of the Maximum Lyapunov Exponent
can be made more clear. It defines $\lambda_{max}$ as the "largest
value" such that equation (6) holds, but it's not clear what the
maximum is being taken with respect to here. From the sentence
preceding definition 3.1, I think there should be a supremum over all
perturbation directions before the limits.

On line 155, the paragraph starts by mentioning Example 4.1 which
the reader hasn't encountered yet. This doesn't read well.

I do not understand the argument about Lipschitz (dis)continuity for
stochastic policies around line 223. It says in the zero-variance
limit that  $\pi_\theta(a\mid s) =
\delta(a - \mu s)$ is
a Dirac measure and cannot be Lipschitz continuous, despite the fact
that the deterministic limit
$\pi_\theta(s) = \mu s$ is Lipschitz continuous, but you are surely
referring to 2 distinct notions of Lipschitz continuity here. The
former (as far as I can tell) is a statement about the Lipschitz
continuity of the map $a\mapsto \pi_\theta(a\mid s)$, whereas the
latter is a statement about the Lipschitz continuity of the map
$s\mapsto\pi_\theta(s)$. These are different things, and it's not
clear to me why it is meaningful to compare them. The fair comparison
as I see it would be the map $s\mapsto \pi_\theta(\cdot\mid s)$ in the
stochastic case, which can indeed be Lipschitz continuous if the image
is metrized by the Wasserstein distance.

On line 89, it says "The sample space $\mathcal{S}$ is
compact". Should it not say that the state space is compact?

Why is $R$ used to denote a cost function? This is confusing.

Line 110, why does the second assumption only "apparently" hold?

On line 118, what does $\simeq$ mean? Is $\|\Delta Z(t)\|\simeq
e^{\lambda t}\|\Delta Z_0\|$ the same as
$\|\Delta Z(t)\| = e^{\lambda t} + o(1)$?

Why is assumption A4 natural? Under which conditions on the MDP/policy
is A4 satisfied?

Other policy gradient works have proposed conditions for which the
policy gradient is defined (see e.g. "Deterministic Policy Gradient
Algorithms" by Silver et. al, 2014; Theorems 1, 2 and conditions A1,
A2, B1).
Is your Theorem 2 related to these conditions? Is it easier to verify
the conditions of Theorem 2?

In the experiments, how exactly was the policy gradient computed? In
particular, how are you estimating the advantage in equation (3)? How
are you computing the policy gradient for deterministic policies?
Also, equation (3) for $\eta$ looks like it is already computing a
gradient w.r.t. parameters, so $\nabla_\theta\eta(\theta_0)$ does not
appear to have the right shape, especially in the expression $\theta =
\theta_0 + \delta\nabla_\theta\eta(\theta_0)$.
I'm assuming by $\nabla_\theta\eta$ you just mean $\eta$.

Are there any promising directions that RL researchers can follow in
order to solve the "fractal landscape" problem?


**Limitations:**

Some assumptions are made for the estimation of the Holder exponent
that seem very restrictive. Also, it is not clear to me if the
condition on the maximum Lyapunov exponent is easier to verify or more
intuitive than those given in earlier works on policy gradient
methods, which also serve the purpose to ensure that the policy
gradient exists.

---

> ### Author Rebuttal · Authors · 2023-08-05
>
> Thank you for the detailed comments and suggestions. Please see the answers below. We should clarify that the results are not meant to be negative (and will revise some of the statements) but to illustrate a source of difficulty in policy optimization that has not been discussed much in existing work, and we believe the results may lead to new ways of improving RL algorithms.
>
> Weakness:
> 1. Yes, we chose the words "very unlikely" because the theory does not exclude the case of using trivial reward functions, such as the constant function $R(s, a)\equiv0$ which always gives a perfectly smooth landscape as you suggested. Stronger results are possible under more restrictive assumptions but the generality will likely be weakened.
> 2. The connection between Holder exponent and the Hausdorff dimension (an important characteristic of fractal) is given in Proposition C.1, which states that the landscape can have fractional dimension if its Holder exponent is less than 1. Importantly, the set of these points is not of measure zero, but determined by the measure of the set $M=(\theta\in\mathbb{R}^N:\lambda(\theta)>-\log\gamma)$. For chaotic systems and the conventional $\gamma=0.99$, the measure of $M$ is positive, indicating the landscape is truly fractal. This claim is validated by Figure 2(a) and 2(d)-2(e).
> 3. Tightness assumption in Holder exponent estimation: really good question. Generally speaking, precisely estimating the Holder exponent of a high-dimensional function is hard, as the function may have different curvatures along different directions. Therefore, our methods aim to estimate whether the objective function $J(\theta)$ is differentiable at given point. The slope is at least 2 when it is differentiable, since it has $|J(\theta')-J(\theta)|\leq K\| \theta'-\theta\|$ for some $K > 0$, and further implies $Var(J(X))\leq K'\sigma^2$. Therefore, one can compare the slope of regression with 2 to determine the smoothness, without worrying about the precise value of the Holder exponent.
>
> Questions:
> 1. Line 57:  We agree that the word "satisfactory" is ambiguous and will further clarify it in the writing. In acrobot swing-up-and-balancing as typically modelled as an RL environment, the default reward function (e.g., in OpenAI Gym) is -1 to all steps that do not reach the target height, and 0 if it achieves the target height, in which case it terminates. This is a weaker version of the original control problem of minimizing the quadratic cost and stabilizing at the top (and the RL-trained policies are "very sensitive" as you pointed out, showing the difficulty in stabilizing).
> 2. The divergence term $\Delta Z(t)=s'(t)-s(t)$ is the difference between two trajectories at time $t$. Regarding the definition of 3.1, yes, it should be given as $$\lambda_{max}=\limsup_{t\rightarrow\infty}\limsup_{|\Delta Z_0|\rightarrow 0}\frac{1}{t}\log\frac{|\Delta Z(t)|}{|\Delta Z_0|}.$$ It does not change the main claims as the limits in the example exist.
> 3. Line 223: For deterministic policies, they are directly from a function approximator which is locally Lipschitz continuous, and we are able to prove Holder continuity only if it is assumed. However, stochastic policies are determined by both mean and variance, and in general it is not Lipschitz w.r.t. the variance (e.g. Dirac function), that is why the landscape can be even more fractal in this case. Therefore, unless further assumptions (e.g., Wasserstein distance as you suggested) are made, the worst case of stochastic policies is worse than that of deterministic policies, as shown in Example 4.3. We believe that it will be a promising research direction, but delving too deeply may deviate our paper from the main thread.
> 4. Line 110, if we assume the first assumption ($\nabla V$ exists and is continuous over $\mathcal{S}$), then the fact that any continuous function over a compact domain is bounded gives the second assumption.
> 5. Line 118, the formula $|\Delta Z(t)|\simeq e^{\lambda t}|\Delta Z_0|$ says that for sufficiently small $|\Delta Z_0|$, there exist $K_2>K_1>0$ such that $K_1e^{\lambda t}|\Delta Z_0|\leq |\Delta Z(t)|\leq K_2 e^{\lambda t}|\Delta Z_0|$ when $t$ is not too large. Furthermore, it is also the motivation of assumption (A.4): the largest exponential divergence rate of trajectories is $\lambda(\theta)$. Positive maximal Lyapunov exponent is an indicator for chaotic systems (see [11]). Thus, a direct implication is that the objective functions in some often-used MDPs (acrobot, hopper) are non-differentiable. In contrast, the landscape of inverted pendulum is smooth as shown in Figure 3(c) since it is not chaotic.
> 6. Regarding the conditions in DPG, the assumptions (underlying MDPs are stochastic) aim to avoid difficulties along the lines of our analysis, but our goal is to show that non-smooth landscapes can happen under very mild assumptions, so typical algorithms are indeed used in MDPs that do not satisfy the assumptions in the DDPG paper, and more work is needed to further improve RL algorithms.
> 7. In the experiments, we use vanilla policy gradient to estimate the gradient from (3). Actually, the policies were stochastic (line 281-285) and hence (3) is applicable. When plotting, we use its deterministic limit to evaluate the objective. In figure 4(d), the landscape is fractal because we are using stochastic policy. However, the landscape in Figure 4(c), 5(c) are also fractal even if everything is deterministic, which implies that non-smoothness is not just a consequence of randomness and thus validates our theory.
> 8. For the future study, given that the landscape is fractal, it will be interesting to consider techniques that can approximate the objective $J(\theta)$ by a smooth surrogate function $J'$, then estimate the gradient of $J'$ as ascent direction. Further interesting questions would then involve how to balance the trade-off between the smoothing effects and the approximation error.

---

> > ### Comment · Reviewer_Qd1b · 2023-08-11
> >
> > > Yes, we chose the words "very unlikely" because the theory does not exclude the case of using trivial reward functions,
> >
> > Are you claiming that the only cases where the small Lyapunov constant is benign are when the reward function is constant? I don't think so, but then the problem still holds: why do you claim that it's very unlikely that the objective will be Lipschitz continuous? Can you be more specific about just how unlikely it is, and how severe the consequences are in that case?
> >
> > > The connection between Holder exponent and the Hausdorff dimension (an important characteristic of fractal) is given in Proposition C.1, which states that the landscape can have fractional dimension if its Holder exponent is less than 1
> >
> > At the very least, this needs to be discussed in more detail in the paper. But still, while you claim that the landscape *can* have fractal dimension less than 1, I would like to see more convincing evidence of the prominence of fractal landscapes in deep RL.
> >
> > > Line 223: For deterministic policies, they are directly from a function approximator which is locally Lipschitz continuous, and we are able to prove Holder continuity only if it is assumed. However, stochastic policies are determined by both mean and variance, and in general it is not Lipschitz w.r.t. the variance (e.g. Dirac function), that is why the landscape can be even more fractal in this case. Therefore, unless further assumptions (e.g., Wasserstein distance as you suggested) are made, the worst case of stochastic policies is worse than that of deterministic policies, as shown in Example 4.3. We believe that it will be a promising research direction, but delving too deeply may deviate our paper from the main thread.
> >
> > I do not agree. Lipschitz continuity is defined w.r.t. metrics on the domain and codomain of a mapping -- it does not make sense to discuss Lipschitz continuity without some notion of metric. The Wasserstein distance suggestion that I made was not an assumption, it was an analytical choice. Your argument about the Lipschitz-discontinuity is also "assuming" a metric on the space of distributions, and I claim that the metric you chose is not very meaningful. For instance, you can take any neural network representing a deterministic policy and then interpret it as a stochastic policy with no variance. By your logic, the Lipschitz continuity of this mapping changes depending on whether you interpret the actions as (deterministic) random variables or simply actions -- even though it is the exact same mapping. The problem of Lipschitz-discontinuity arises due to a choice of metric that you implicitly made. My argument is that the Wasserstein metric is a more meaningful one -- when talking about Lipschitz continuity w.r.t. the Wasserstein metric, in my previous example, the policy would be simply be Lipschitz continuous, regardless if you look at the outputs as random variables (which happen to be deterministic) or actions.
> >
> > > Line 110, if we assume the first assumption...
> >
> > Ok, that makes sense, but I don't think the sentence should start with "Apparently". It sounds like as if someone told you that the assumption holds, and you're just taking their word for it :)
> >
> > Maybe this is just a consequence of how I'm used to hearing the word. For some reason, replacing "Apparently" with "It is apparent that" seems more appropriate, even though I suppose they technically mean the same thing.

---

> > > ### Author Response · Authors · 2023-08-12
> > > **Response**
> > >
> > > Thanks for the timely response, and we are happy to answer your follow-up questions.
> > >
> > > 1. "Are you claiming that the only cases where the small Lyapunov constant is benign are when the reward function is constant?"
> > >
> > > A: The objective function is smooth if the Lyapunov exponent $\lambda(\theta)$ is small as it increases the Holder exponent $\frac{- \log \gamma}{\lambda(\theta)}$, and we suppose that you meant "large Lyapunov exponent". The claim we tried to made is that under the current assumptions, it has
> > >
> > >   (a) The objective function $J(\cdot)$ is $\frac{- \log \gamma}{\lambda(\theta)}$-Holder continuous at $\theta$;
> > >
> > >   (b) A counterexample is provided that it is also the strongest Holder continuity that can be proved.
> > >
> > > Therefore, we are not trying to say that only trivial rewards work in the case of large $\lambda(\theta)$. Since we only assumed that the reward function is Locally Lipschitz, it does not prohibit one from using constant rewards. Regarding how ``unlikely" it is, recall that the reason why chaotic systems have fractal landscapes is their exponential divergence of nearby trajectories. Therefore, as long as a reward function can continuously distinguish two separate trajectories (e.g. $|x(t)|$, $x^T(t) Q x(t)$), the objective function will be non-smooth, as shown in experiments.
> > >
> > > 2. We agree that more coverage of fractal theory should be moved to the main text and we will revise our paper accordingly.
> > >
> > > Regarding the fractal landscapes in RL, since there are very few previous works on analyzing the landscape in RL other than finite spaces and linear MDPs as we discussed in the Related work, it is hard to find any paper that directly points out this phenomenon. A direct evidence is that in the acrobot and hopper tasks, the slope of linear regression is significantly less than 2, which suggests lower Holder exponents there. Actually, all of us have observed fluctuating training curves, which is sometimes a reflection of the underlying fractal landscape. Aside from examples provided in our paper, we can see that in the PPO paper, ([22]) the training curves in chaotic MDPs (hopper, inverted double pendulum, etc.) fluctuate much more than those in non-chaotic systems (reacher, swimmer, etc.).
> > >
> > > Just one more thing we would like to clarify: although many fractals (e.g., image of the Weierstrass function, the Cantor set) have self-similar structures, a set is not required to be self-similar for being a fractal. Actually, the core feature of fractals is their fractional Hausdorff dimension which is closely related to the Holder exponent of the underlying mapping (through Proposition C.1). Therefore, the landscapes we showed in this paper are truly fractals even if they do not exhibit a strong self-similarity.
> > >
> > > 3. Regarding the transition from deterministic policies to stochastic policies, we agree that Wasserstein distance can be a good metric to use. The point we were trying to make is that unless additional assumptions are made, there are some stochastic policies that can lead to a landscape as non-smooth as those by deterministic policies.
> > >
> > > Consider Example 4.3 with $\beta = 1$. The 1-Wasserstein distance between the distributions $\delta(x - |\theta_1|s)$ and $\mathcal{U}( |\theta_1| s +  |\theta_2|, |\theta_1| s + 2 |\theta_2|)$ is $W_1(\theta_1, \theta_2) = \frac{3 |\theta_2|}{2}$ by definition. Therefore, the policy is Lipschitz continuous w.r.t. the 1-Wasserstein metric at $\theta_2 = 0$. However, we have proved the Holder exponent of $J(\theta)$ in Example 4.3 is again $\frac{-\log \gamma}{\lambda(\theta)} < 1$, which means that the objective function can still be non-smooth even if the policy is Lipschitz continuous w.r.t. the Wasserstein metric. Therefore, it supports our claim that more assumptions (e.g., the probability density function is point-wise smooth) are necessary to rule out this case. We believe that it is clearer to demonstrate with the aid of Wasserstein distance, and will incorporate our discussion into the paper.
> > >
> > > 4. Thank you, we will change the writing.

---

> > > > ### Comment · Reviewer_Qd1b · 2023-08-14
> > > >
> > > > > we suppose that you meant "large Lyapunov exponent"
> > > >
> > > > Yes, my apologies.
> > > >
> > > > > A counterexample is provided that it is also the strongest Holder continuity that can be proved.
> > > >
> > > > My criticism is that there appears to be a gap between "we cannot prove that the objective will be at least $\alpha$-Holder continuous" and "the landscape will be fractal". Reviewer dey2 seems to have the same concern, and maybe they stated it better in their section "Disconnect between the stated result and the non-smoothness claim". In your rebuttal, you say "we demonstrate that the objective function can be non-smooth", and I agree with that statement. What still remains unclear to me is whether the maximum Lyapunov coefficient is actually  a viable test of whether or not a policy gradient algorithm will work in a given environment. Particularly, I suspect the test proposed in the paper would produce many "false negatives". Right now, this test consists of estimating the Holder coefficient of the objective, and returning "objective is smooth" when a threshold is crossed. What does the test say otherwise? My understanding is that the test says "we do not know whether or not the objective is smooth", and it is not clear to me how much we should really be biased to believe that the object is non-smooth in this case.
> > > >
> > > > > which means that the objective function can still be non-smooth even if the policy is Lipschitz continuous w.r.t. the Wasserstein metric. Therefore, it supports our claim that more assumptions (e.g., the probability density function is point-wise smooth) are necessary to rule out this case
> > > >
> > > > Agreed. To be clear, I did not mean to imply that stochastic policies would outright solve the problem, but anyway it is interesting to see for sure that they do not. I was mostly critiquing the statement "... which further implies that the loss landscape of J(·) can become even more fractal and non-smooth in the case of stochastic policies, especially when it approaches its deterministic limit", which I do not think is quite right.

---

> > > > > ### Author Response · Authors · 2023-08-14
> > > > > **Response**
> > > > >
> > > > > Thanks for the comments.
> > > > >
> > > > > 1. "whether the maximum Lyapunov coefficient is actually a viable test of whether or not a policy gradient algorithm will work in a given environment.": There is one thing that we would like to clarify: in this paper, we only claimed that maximal Lyapunov exponents can be used to determine whether the landscape around certain points is smooth, and policy gradient methods are not well-posed when the landscape is fractal, because there is no gradient to estimate at all. Actually, being ill-posed does not imply that an algorithm fails to work entirely. Instead, it indicates that the algorithm relies on certain assumptions that are not met. Specifically, there could be some special combinations of step size and search direction for the policy gradient that yield "right" updates. However, there are two serious issues:
> > > > >
> > > > >    (a) In practice, there is no way to know beforehand what the correct direction and step size are, or even if they exist;
> > > > >
> > > > >    (b) The impact of fractal behavior significantly grows as the step size decreases. However, most policy gradient methods (e.g., TRPO, PPO) suggest that the step size must be small to ensure policy improvement.
> > > > >
> > > > >    Therefore, our work is not intended to "attack" existing methods in any sense. Instead, we try to elaborate a phenomenon which has not yet received much attention, but can cause serious problems to the fundamentals of all gradient-based methods. We will be pleased to see if any approaches are proposed to tackle this problem in the future.
> > > > >
> > > > > 2. "I suspect the test proposed in the paper would produce many 'false negatives'": Since we have shown that when the objective function is locally Lipschitz continuous, the slope of linear regression is no less than 2, it implies that the landscape is indeed non-smooth if the slope of regression is significantly less than 2 as observed in acrobot and hopper tasks. Here we used the fact that a claim is true if its contrapositive claim is true, and hence no false negative will be produced.
> > > > >
> > > > > 3. "Agreed. To be clear, I did not mean to imply that stochastic policies would outright solve the problem, but anyway it is interesting to see for sure that they do not.": Thank you, we will improve the writing based on our discussion.
> > > > >
> > > > > We hope that this response addresses your concern. We would greatly appreciate it if you could consider updating the score.

---

### Official Review · Reviewer_bNWE · 2023-06-19

**Soundness:** 4 excellent
**Presentation:** 3 good
**Contribution:** 4 excellent
**Rating:** 9
**Confidence:** 4

**Summary:**

The paper points out that the value function of some MDPs is not differentiable or even fractal in the infinite horizon setting.

This raises the question of the well-formedness of policy gradient methods.

A method for statistically estimating the Hölder exponent is presented to find out whether the training process has encountered fractal landscapes.

**Strengths:**

* The paper describes a fundamental problem that has received little or no attention to date.
* The presentation is well-founded and provides deep insights.
* The Appendix is a very nice addition to the main text.

Remark:\
I observed the presumably fractal structure of the Q-function of an MDP myself in 2017. However, I decided against elaborating and publishing this observation further at that time. I would like to congratulate the authors for their sound and profound elaboration of the description of this phenomenon.

**Weaknesses:**

I see no weakness.

Minor notes:

Some of the capitalization is inconsistent "Hölder Exponent", "Hölder exponent". \
I suggest:\
“Maximum Lyapunov Exponents” -> “maximum Lyapunov exponent”\
„Hölder Exponent“ -> „Hölder exponent“

There are a couple of typos:\
„updare” -> “update”\
“even small update” -> “even small updates”\
„transitons” -> „transitions”\
“pertubation” -> “perturbation”\
“issues[27]” -> “issues [27]”

The sentence “Similar result obtained for the H2/H∞  problem [33].” is incomplete. Perhaps it should read “Similar results are obtained for the H2/H∞  problem [33].”

To make the use of hyphens consistent, it should be "non-linear" instead of " nonlinear", analogous to "non-smooth".


In Subsection 4.1 and 4.2, the mathematical expressions should be in bold, e.g. with boldmath


In the References there are still some unintentional, wrong lowercase letters:\
goldstein\
lyapunov

In “Since these features are not necessarily binded to certain types of continuous state-space MDPs”, instead of "binded" it should rather be "bound" or "tied".

**Questions:**

Would you agree with the view that for stochastic MDPs where the probability distribution of the subsequent states P(s'|s,a) is reasonably smooth and a continuum of subsequent states is possible (probably the condition is that the distribution function is differentiable) there can be no more fractal value function? The probability distribution acts like a smoothing kernel and makes everything differentiable.

**Limitations:**

Yes

---

> ### Author Rebuttal · Authors · 2023-08-05
>
> Thank you very much! We are delighted that you are interested in our paper!
>
> The question you raised is very insightful, and we would like to share some opinions about it.
>
> In Example 4.3, we showed that if no additional assumptions are added, using stochastic policies may lead to a more non-smooth landscape than that created by deterministic policies when the underlying MDP is deterministic. In the DPG work [23], the authors proved that if the underlying MDP is stochastic and satisfies certain conditions (including the one that you suggested in the question), then the objective function is differentiable even when using a deterministic policy.
>
> Intuitively, the randomness in stochastic MDPs may create a smoother landscape, as the probability distribution may act like a smoothing kernel. A similar result is in the search gradient method ("Natural Evolution Strategies" by Wierstra et al.), where it smooths the objective function $J$ by convolving it with a Gaussian kernel $p(\mu, \sigma)$ so that the new objective $J * p$ (convolution of $J$ and $p$) is smooth w.r.t. $\mu$ and $\sigma$," allowing gradient descent algorithms. Therefore, we totally agree with your idea.
>
> Just one more thing that we want to mention: one of the main continuous-space environments for RL is control and robotics. If we ignore the error and noise in those environments, the underlying models are all deterministic as they are derived from physical laws. Therefore, a dilemma is posed to us: if we have much noise in the system, the objective function may be smoother (assuming the noise satisfies all conditions in [23]). However, in the meantime, the system will be less stable and controllable. On the other hand, if we have little noise, the objective function will be fractal and non-smooth, which makes gradient-based methods less effective.
>
> Overall, we believe that it is a very important and fundamental problem in RL that deserves more attention in the future.

---

### Official Review · Reviewer_xgpg · 2023-07-05

**Soundness:** 4 excellent
**Presentation:** 4 excellent
**Contribution:** 4 excellent
**Rating:** 7
**Confidence:** 4

**Summary:**

This paper proposes a method to understand the optimization landscape of the policy gradient method in the search for a non-smooth or fractal landscape. The method is based on the chaos theory and non-smooth analysis of the optimization objective. In particular, the paper proposed a method to estimate the Hölder exponents, which is used to identify non-smooth landscapes. The theoretical analysis is discussed with empirical validation.

**Strengths:**

- The paper tackles an important and fundamental problem in policy gradient.
- Overall a well-written paper with a comprehensive presentation.
- The theoretical results are explained well with necessary assumptions and analysis.
- Empirical results are presented in support of the theoretical analysis.

**Weaknesses:**

- Some experimental details (algorithm, return curve, random seed) are missing.
- Unclear how the proposed method can be integrated into various effective policy gradient methods (e.g., Vanilla Policy Gradient, PPO).

**Questions:**

It is unclear what the conclusion of the analysis on the non-smoothness of stochastic policies is in Section 4.3. Are the stochastic policies preferred over deterministic ones if the MDP (task) has a non-smooth landscape?

Some experimental details are missing. The analysis was conducted on several Gym environments; however, the learning curve and RL algorithm details are absent. What algorithm was used for policy learning? Was it a vanilla policy gradient? Was batching employed for parameter updates, or were they updated after every sample? Providing a detailed algorithm would address these questions.

How the analysis depends on any particular type of policy gradient implementation, like with a batch of the sample or a single sample for parameter update?

To assess the algorithm's performance, it is crucial to examine its reward. Therefore, in addition to the loss curve, a return curve would provide a better understanding of the challenges involved in solving a complex task with a non-smooth landscape.

Acrobot is mentioned as (lines 301-302) almost unsolvable by the policy gradient method. However, many policy gradient-based algorithms, such as PPO, can already solve this task. See CleanRL benchmark results for PPO (https://github.com/vwxyzjn/cleanrl). Is the analysis in this paper restricted to some particular policy gradient-based methods?

Is the experiment run for multiple random seeds? The execution of the environments (MDP) changes in each episode; thus, multiple runs would give a better understanding. Adding multiple seed runs or explaining why that is unnecessary is crucial.

The Axis labels of Figures 2, 3, 4, and 5 are too small and hard to read.

**Limitations:**

Yes.

---

> ### Author Rebuttal · Authors · 2023-08-05
>
> Thank you for your positive feedback. We're glad to hear that you liked our paper!
>
> $\textbf{The conclusion in Section 4.3}$: We claimed that unless further assumptions are made, the worst case of stochastic policies is worse than that of deterministic policies, as shown in Example 4.3. Therefore, if we would like to solve the deterministic MDP using stochastic policies, we must be careful when choosing any specific probability distribution. We believe that it will be a promising research direction, but delving too deeply may deviate our paper from the main thread. It would be nice to see if any progress can be made in the future.
>
> $\textbf{Integration with existing algorithms}$: We believe that it will be a promising direction for future research. In Section 5, we proposed a statistical method that determines whether a function is differentiable at a given point, which can be used to check if the training has encountered a fractal landscape. If such a scenario arises, it indicates that further measures (local smoothing, re-initialization, etc.) are required and thus the objective function can be optimized efficiently. Also, the fact that the landscape can be fractal provides novel insights for practitioners to design more effective algorithms.
>
> $\textbf{Numerical experiments}$: Since most concerns are concentrated on the experiments, we would like to address your questions from the following aspects:
>
> 1. The purpose of experiments: The numerical experiments are intended to serve as validations for the theoretical results, and the thing we are interested in is the non-smoothness of the landscape around a randomly sampled $\theta_0$. Therefore, as long as we can observe a clear fractal behavior in the plots, it will be in support of our theory.
>
> 2. How they are conducted: In each numerical task, we do the following three things:
>
>     (a) Randomly sample $\theta_0$ from a Gaussian distribution;
>
>     (b) Perform one-step vanilla policy gradient to estimate a descent direction $\eta$ at $\theta_0$;
>
>     (c) Evaluate the loss $J(\theta_0 + \eta \delta)$ along $\eta$ for different $\delta$, and observe the non-smoothness.
>
>     Therefore, it is not a training process. The central claim is that when the landscape is fractal, there is no descent direction to estimate at all, which makes all policy gradient methods ill-posed. But we agree that adding more experimental details will make the paper more self-contained.
>
> 3. The RL algorithm: We used Vanilla policy gradient to estimate the gradient of the advantage function (the original objective $J$ can be non-differentiable). Since what we want to show is the non-smoothness of the landscape, we are not training the policy, nor comparing different algorithms. Suppose that we have an objective $J(\theta)$ which is non-differentiable at $\theta = \theta_0$. Then all policy gradient methods fail to estimate its gradient at $\theta_0$ accurately regardless of what the algorithm actually is, because there does not exist a gradient at all in the first place. In the experiments, we showed that Vanilla PG cannot find a descent direction in chaotic systems (acrobot, hopper), and we believe that similar results can be obtained using other variants of policy gradient methods. In a word, having fractal landscape is an intrinsic property of chaotic MDPs, and is independent of how we solve it. We will make it clearer when updating the paper.
>
>
> 4. The hyper-parameters: Since it is only a one-step policy gradient algorithm, the following parameters should be sufficient to reproduce the results along with those provided in the paper:
>
>     The random seed for vanilla policy gradient: np.random.seed(42);
>
>     Minibatch = 16;
>
>     Horizon = 1000;
>
> 5. Eliminating the randomness: In experiments, the policies we selected are stochastic (Gaussian distribution) so that vanilla PG is applicable. However, when plotting the loss curve $J(\theta_0 + \eta \delta)$ along $\eta$ for different $\delta$, the problem is that if we use the stochastic policy directly, the value of objective function $J$ has to be estimated by sampling the Gaussian random variables in the policy. Since paths of Gaussian noise are recognized for their fractal nature due to inherent randomness, it will undoubtedly interfere the desired observations. The reason is that we cannot tell whether the fractal landscape is a consequence of randomness or the intrinsic chaotic behavior within the MDP. Therefore, we apply the deterministic limit of the policy (by setting variance to 0) when evaluating $J(\theta)$ to eliminate randomness. As expected, the landscapes are fractal in chaotic systems (acrobot, hopper) even when everything is deterministic, and smooth in non-chaotic systems (inverted pendulum).
>
> $\textbf{The solvability of acrobot}$: We agree that the word "unsolvable" might be ambiguous and will further clarify it in the writing. In our paper, when we refer to the acrobot task, we specifically mean the swing-up-and-balancing task, not just the swing-up part itself. To the best of our knowledge, there is still a lack of effective policy gradient algorithms that can achieve this task, even though control methods have successfully solved it many years ago ("The Swing Up Control Problem For The Acrobot" by Spong, 1995). As for the mentioned algorithm "cleanRL," we noticed that it uses the default reward function from OpenAI Gym, which assigns -1 to all steps that do not reach the target height and assigns 0 if it achieves the target height and terminates. However, typically, a quadratic cost term like $x^T Q x$ is necessary for balancing. Therefore, using such a reward function can only swing up the acrobot and cannot stabilize it to the upright position.
>
> Hope that the above responses have addressed your concerns, and we are happy to answer any further questions.

---

> > ### Comment · Reviewer_xgpg · 2023-08-19
> >
> > I would like to thank the authors for their detailed responses. Many of my concerns have been addressed. However, demonstrating how the landscape influences the results, especially the reward performance, would further improve the paper. That would emphasize the importance of identifying the (fractal) landscape. Thus, I have raised my rating from 6 to 7.

---

> > > ### Author Response · Authors · 2023-08-21
> > > **Thanks**
> > >
> > > We sincerely appreciate your support for our work! Adding a discussion on how the reward performance is affected by the fractal landscape can convey a stronger message to practitioners in RL, and we will try to incorporate it into the main text or appendix.

---

### Official Review · Reviewer_dey2 · 2023-07-06

**Soundness:** 2 fair
**Presentation:** 2 fair
**Contribution:** 3 good
**Rating:** 4
**Confidence:** 3

**Summary:**

The paper tries to argue that in a certain class of continuous state-action MDPs, the policy gradient landscape can be non-smooth. It does so by showing that the Holder exponent of the policy objective $J(\theta)$ is less than one, when the MDP's maximum Lyapunov exponent is greater than $-\log \gamma$. The paper complements this result by analytically analyzing multiple simple MDPs and also by empirically showing this non-smooth behavior for popular MDPs like Acrobot.

Unfortunately, I propose rejecting this paper because I am not confident in its proofs and cannot follow the logic completely. Although, it is possible that I am missing something, in which case I hope that the author's rebuttal can help alleviate my concerns (please see the Weaknesses and Questions sections below.)

**Strengths:**

### Originality and significance:
The paper deals with the fundamental issue of characterizing the nature of the policy optimization landscape in continuous state-action MDPs. This setting is highly practical, and to the best of my knowledge, not widely studied. One of the primary advantage of policy gradient methods over value based methods is their ability to naturally deal with continuous action-spaces. And as the paper claims, the optimization landscape is non-smooth which might create significant difficulties for gradient based methods. This knowledge is very useful for both algorithm developers and practitioners. The approach itself seems elementary and novel to me (although I have very limited knowledge about related works). (Also, I mean "elementary" in a good way. This seems like those first fundamental results which other people build upon.)

### Quality and clarity:
The paper motivates the problem well and provides a thorough literature survey, with an introductory background on chaos and fractal theory included in the appendix, which I think would be appreciated by the readers new to these fields. The main analytical results (Holder exponent of the policy gradient objective being less than one) are complemented by a number of worked out examples (simple MDPs) and corroboration by multiple simulation studies (visualizing the landscape of popular environments such as Acrobot and Humanoid showing the non-smooth nature of the landscape), providing a well-rounded treatment of the subject matter.

**Weaknesses:**

## Major issues (these affect my score significantly):

The major weaknesses of the paper are a lack of a coherent argument and flaws in the proofs.

### 1. Disconnect between the stated result and the non-smoothness claim.
The main results (Theorem 4.2, line 192) says that the policy gradient objective $J$ is Holder with some exponent less than one. And the paper then concludes that $J$ is non-smooth in the policy parameters. I don't completely understand this connection. I think that if a function is $\alpha$-Holder for some $0 < \alpha < 1$ and is not Holder continuous of any other $\alpha' > \alpha$, i.e. in some sense $\alpha$ is "max Holder coefficient" (not sure what the terminology is), then it would be non-smooth (can the authors provide the relevant result here as well). However, in the paper, the main result just shows that $J$ is $\alpha$-Holder for some $\alpha < 1$; it doesn't show that this is the maximum possible Holder coefficient. There is a discussion following Theorem 4.1, from line 152-177 that argues that the Holder coefficient for the value function should be less than one. However, if that is a result, it should be the main theorem statement (it does seem very important). The non-smoothness could then follow from the bound on the Holder coefficient, and the theorem (which the authors should state) that connects this max Holder coefficient to the non-smoothness of the function.

Further, from the experimental results (Section 6), all the figures plot $J$ vs $\delta$. But how that is connected to the non-smoothness of the landscape is a little bit unclear. More explanation would be very helpful.

The conclusion of the paper says "It also poses a serious question to the well-posedness of policy gradient methods given the fact that no gradient exists in many  continuous state-space RL problems." But I am unable to attribute this to a specific theorem or empirical result given in the paper.

### 2. Potential flaws in the proofs.
I should state that it is quite possible that these are not really flaws, and I am just unable to follow the proofs. In that case, I would appreciate if the authors could help me here. It is also possible that these are just local problems, and the veracity of the final claims remains valid.

**(a) Proof of Theorem 4.1 (line 455, page 13):**

In the display following Eq. 18, I don't follow the inequality (ignoring the term $K_1K_2 \delta$)

$$ \sum_{t=0}^{T_1} e^{(\lambda(\theta) + \log \gamma)t} \leq \frac{e^{\frac{\lambda(\theta) + \log \gamma}{\lambda(\theta)} \log \left( \frac{2M}{K_1 \delta} \right) + 1}}{e^{\lambda(\theta) + \log \gamma} - 1}. $$

What I am able to compute is the following. Since $\lambda(\theta) + \log \gamma > 0$, the function $e^{(\lambda(\theta) + \log \gamma)x}$ is increasing. Therefore, using the fact that $T_1 \geq \frac{\log(2M / K_1 \delta)}{\lambda(\theta)}$, we get

$$ \sum_{t=0}^{T_1} e^{(\lambda(\theta) + \log \gamma)t} = \frac{e^{(\lambda(\theta) + \log \gamma) (T_1 + 1)} - 1}{e^{(\lambda(\theta) + \log \gamma)} - 1} \geq \frac{e^{\frac{(\lambda(\theta) + \log \gamma)}{\lambda(\theta)} \left[ \log \left( \frac{2M}{K_1 \delta} \right) + \lambda(\theta) \right] - 1}}{e^{(\lambda(\theta) + \log \gamma)} - 1}.$$

**The most important thing to note is the inequality sign is flipped.** Maybe this could be fixed using $T_1 \leq \frac{\log(2M / K_1 \delta)}{\lambda(\theta)}$, but then could we guarantee that $T_1$ would exist?

Similarly, I don't understand the inequality sign in the second big display on page 13: $\frac{\gamma^{T_1}}{1 - \gamma} \leq \frac{1}{1 - \gamma} (K_1 / 2M)^{-\log \gamma / \lambda(\theta)} \delta^{-\log \gamma / \lambda(\theta)}$.

Finally, I don't understand how the Holder continuity result follows from the last equation on page 13. Seems like one step is missing on line 464.

**(b) Proof of Theorem 4.2 (line 466, page 14):**
I don't understand the third line in the first display. In particular, how to show that
$$ R(s_t'; u'(s_t')) + \gamma V^{\pi_\theta}(s_{t+1}') = Q^\theta (s_t', u'(s_t')).$$

Using Bellman equation (for deterministic dynamics), we know that $Q^\theta (s_t', u'(s_t')) = R(s_t'; u'(s_t')) + \gamma V^{\pi_\theta} ( f(s_t', \pi_\theta(s_t'))$. But is it necessary that $f(s_t', \pi_\theta(s_t') = s_{t+1}') = f(s_t', \pi_{\theta'}(s_t'))$?

**(c) (MINOR) Proof of Theorem 4.3 (line 471, page 14):**
I am unable to follow the second line and the fourth line in the first display. (I guess, I just need to think more carefully for these..)

**(d) (MINOR) Example 4.1 (line 183, page 5):**
I don't understand why $||\Delta Z(t)|| = ||\Delta Z_0|| \theta^t$? I do understand that $||\Delta Z(1)|| = | f(s_0', \theta s_0') |$, but is $f(s_0', \theta s_0') = \theta s_0'$?

Also, I don't understand if we get the result $V(\delta) - V(0) \geq \frac{\gamma}{1 - \gamma} \delta^{(-\log \gamma / \log \theta)}$, why does it mean that $p = -\log \gamma / \log \theta$ is the largest Holder coefficient? For instance, we know that $V(\delta) - V(0) = V(\delta) = \delta \sum_{t=0}^\infty (\gamma \theta)^t = \frac{\delta}{1 - \gamma \theta} \geq \frac{\delta^2}{1 - \gamma \theta}$, for small $\delta$ and $\theta$. Does that mean $p = 2$ is the largest Holder coefficient? Some additional explanation would be very helpful.

### 3. There is almost no exposition on how the policy optimization landscape has fractal properties.
There is an appendix that discusses fractal theory. However, since "fractal" appears in the title of the paper, these details (IMHO) shouldn't be relegated to the appendix. In particular, after skimming that appendix, it seems that the discussion there is on general fractal theory, and no explicit connection to the paper's main results (Holder exponent and non-smoothness of the objective function) is made. The main question that remains in my mind is how is the non-smoothness of the landscape connected to it being fractal? (BTW, I think the non-smoothness result in itself is of sufficient interest, and the fractal part could be safely dropped.)



## Minor issues / suggestions (these do not affect my score as much; please ignore them if you don't agree):
- line 15: References 17, 24 are not about policy gradient methods
- line 19 (the probability of obtaining a satisfactory solution can be surprisingly low, or even close to zero in certain tasks such as acrobot): a reference would be nice. I remember the following paper talks about PG methods failing with non-zero probability (maybe that's a little bit relevant?): Mei et al. (2021) Understanding the Effect of Stochasticity in Policy Optimization. NeurIPS.
- line 25 (Considering that even a small update in policy space can significantly change the performance metric): a reference here would be nice
- lines 27-38: some symbols ($\gamma$, $\sigma$) or choices (such as the policy parameterization) are not specified
- line 39 (As shown in Section 6, the loss curve is still fractal even when all things in the MDP are deterministic.): I could not find this result. Could you refer to the exact theorem or the experimental result?
- line 58: Is the edge of stability work a relevant reference for this section? Such as Arora et al. (2022) Understanding Gradient Descent on Edge of Stability in Deep Learning?
- line 73 (It also explains why the classical RL algorithms in [25] are provably efficient in finite space settings.): some references would be nice
- lines 105: for this equation, it would be nice if you could point to the exact section of the reference [26]
- line 110 (However, as we will see in Section 4 and 6, the existence of $\nabla V$ may fail in many continuous MDPs even if $\mathcal{S}$ is compact): I don't see this exact result anywhere. Could you please refer to the specific theorem or the experimental result?
- lines 116-128: could you explicitly define $\Delta Z_t$
- line 128 (but is applicable to a broad class of continuous state-space MDPs having any one of these characteristics) what characteristics is the paper referring to?
- line 133 (Definition 4.1) Should this be called local Holder continuity?
- regarding the Assumptions (lines 146-148 / 93-97): it would be nice if they are written in the same style as a definition or theorem and you can reference to them in the paper.
- line 141 (that the MLE of (1), say λ(θ)): I think the system here is not that given by Eq. 1, but rather $s_{t+1} = f(s_t, \pi(s_t))$
- line 146 (Assumption 4): This is assumption is the most non-trivial one and very critical to the analysis. Some explanation as to why this is reasonable could be useful.
- None of the appendices are referenced in the main paper. Further, Appendices A and C seem highly tangential to the main paper, especially because chaos and fractal theory are never explicitly discussed.
- line 154: this sentence is awkward, has a forward reference to Example 4.1, and doesn't describe that the following text will discuss a lower bound. An appropriate section heading here would be very useful.
- lines 162-164: define $T_i$s
- line 164 (Eq. 8): I am unable to get the factor of $e^{(\log \gamma) T_2}$ in the inequality.
- line 176: should the inequality be (sign reversed): $(p-1)/(\lambda(\theta) + \log \gamma) \geq p / \log \gamma$?
- line 184: I don't understand how to get the equation following this line. Some more explanation could be helpful..
- line 195 (Actually, the most important implication of this theorem is that $J(\theta)$ is not guaranteed **(and is very unlikely)** to be Lipschitz continuous when): why is it very unlikely? Is this an analytical result, or an empirical observation? In either case, some references would be nice.
- Figure 2-5: the figures and especially their font is extremely small to be read when the paper would be printed
- line 213: define $\omega, \Omega$
- line 214 (sending the variance to 0): variance of what, the policy?
- Section 6: all the sub-figures should have axis labels. Also the figure captions could be more descriptive.
- The text has some minor grammar / typographical issues which should be easy to fix. For instance, line 97 spelling of transitons, or line 133 (definition 4.1): suppose that some scalar $\alpha > 0$. --- this sentence seems grammatically incorrect.

**Questions:**

I would really appreciate if the authors could clarify the following (please refer to the Weaknesses section for details):

(1) What is the connection between Holder coefficient and the non-smoothness of the objective? And don't we need a bound on the "max Holder coefficient" for getting the non-smoothness result?

(2a) Is this an actual flaw in the proof, or am I missing something?

(2b) Is this an actual flaw in the proof, or am I missing something?

(2c) Could you please provide more details?

(2d) Am I missing something?

(3) Does non-smoothness of the objective function directly imply that it is fractal in nature?

**Limitations:**

The paper does not comprehensively discuss the limitations of the project.

Assumptions, in particular A4, could have more motivation.

The case when $\lambda(\theta) < -\log \gamma$ could be further discussed; at the very least some comment on what the authors think would happen in that case could be helpful. (Maybe this reasoning is trivial, but I'll still encourage the authors to add that.)

---

> ### Author Rebuttal · Authors · 2023-08-05
>
> We appreciate the reviewer for the detailed comments and questions. We focus on addressing the major issues, and will further improve the writing based on the list of minor issues.
>
> 1. The main result in this paper is that given a policy parameter $\theta$, the objective function $J(\theta)$ is $\frac{- \log \gamma}{\lambda}$-Holder continuous when the maximum lyapunov exponent $\lambda(\theta) > -\log \gamma$. The result itself does not rule out the possibility of having stronger Holder continuity, so we also provided an example (Example 4.1) to show that there exists an MDP that satisfies all assumptions (A.1)-(A.5), but its objective function is not $p$-Holder continuous for any $p > \frac{-\log \gamma}{\lambda(\theta)}$. A discussion and Example 4.1 following Theorem 4.1 provided further support.
>
> A continuously differentiable function is always Lipschitz (Holder exponent $\alpha = 1$). Having smaller Holder exponents usually indicates that a function is not differentiable. Therefore, by showing that the largest Holder exponent $\frac{-\log \gamma}{\lambda(\theta)}$ that be guaranteed is less than 1, we demonstrate that the objective function can be non-smooth. Our theory does not prohibit one from using trivial reward functions, such as $R(s, a) \equiv 0$, which gives a constant objective whatever the MDP is. We have focused on giving the general results, instead of imposing more restrictive assumptions for ruling out different types of trivial reward functions.
>
> For the conclusion, since all gradient-based methods work in the following way: estimate an ascent/descent direction $\eta$ at given policy $\theta_0$, then update the policy $\theta' = \theta_0 + \eta \delta$ along that direction. Therefore, a necessary condition is that $J(\theta)$ must allow a linear approximation (and hence be Lipschitz) in a neighborhood of $\theta_0$, otherwise there does not exist a monotonically increasing/decreasing direction at all. We provided theoretical and experimental results to demonstrate that $J(\theta)$ may be non-smooth. The results are not necessarily negative, as they may open up new ways of further improving RL algorithms.
>
> 2. For the questions in proofs:
>
> (a) In fact, we assumed that $T_1$ is the smallest integer that satisfies (18) (line 457-458), thus we have $\frac{1}{\lambda(\theta)} \log (\frac{2M_2}{K_1 \delta}) \leq T_1 \leq \frac{1}{\lambda(\theta)} \log (\frac{2M_2}{K_1 \delta}) + 1$. Therefore, it should have no sign-flipping issues. There is an error on the constant factor in our first calculation, now it is fixed as
> $$|V^{\pi_\theta}(s'(0)) - V^{\pi_\theta}(s(0))| \leq (\frac{e^{2( \lambda(\theta) + \log \gamma)} K_2 K_1^{\frac{- \log \gamma}{\lambda(\theta)}} (2M_2)^{1 + \frac{\log \gamma}{\lambda(\theta)}} }{e^{(\lambda(\theta) + \log \gamma)} - 1 } + \frac{2 M_2}{1 - \gamma} (\frac{K_1}{2 M_2})^{\frac{-\log \gamma}{\lambda(\theta)}}) \  \delta^{\frac{-\log \gamma}{\lambda(\theta)}}$$
> which also implies the $\frac{-\log \gamma}{\lambda(\theta)}$-Holder continuity as claimed.
>
> (b) Actually, $\{s'(t)\}$ is the trajectory generated by policy $\pi_{\theta'}$, therefore it has $f(s'(t)), \pi_{\theta'}(s'(t))) = s'(t+1)$ and we did not use $f(s'(t), \pi_{\theta'}(s'(t))) = f(s'(t), \pi_{\theta}(s'(t)))$. Regarding the definition of Q-function, we use the one from [25], page 78 which gives $Q^\pi(s, a)=R(s, a)+\gamma V^\pi(f(s, a))$ in the deterministic case.
>
> Indeed, Lemma 4.1 is an existing result and we don't take any credit for its derivation. The undiscounted version of this lemma can be found in the appendices of [7] by Fazel et al. (Lemma 10).
>
> (c) We use the following facts here: (a) $R(s, a)$ is locally Lipschitz w.r.t. $a$; (b) when $\|a' - a\|$ is small enough and $0< \alpha < 1$, it is bounded above by $\|a' - a\|^\alpha$.
>
> (d) For Example 4.1, consider a different initial state $s'(0) = \delta > 0$, by employing the policy $\pi_\theta(s) = \theta s$, the MDP becomes $s'(t+1) = f(s'(t)), \pi_\theta(s'(t))) = \pi_\theta(s'(t)) = \theta s'(t) = \theta^t \delta$ when $\delta \theta^t \leq 1$. If $s_0 = 0$, then $s(t) = 0$ for all $t$. Therefore, $\| \Delta Z(t) \| = |s'(t) - 0 | = \delta \theta^t$.
>
> For the maximality of Holder exponent, suppose that there exists $p > \alpha$ (denote $\alpha = \frac{-\log \gamma}{\log \theta}$ for simplicity) such that $V(s)$ is $p$-Holder continuous at $s = 0$. Then, we can find some constants $M > 0$ and $d > 0$ such that $|V(\delta) - V(0)| \leq M \delta^p$ for all $|\delta|<d$ by Definition 4.1. However, we have already had $|V(\delta) - V(0)| \geq \frac{\gamma}{1 - \gamma} \delta^\alpha$ for all sufficiently small $\delta>0$, combining them yields $\frac{\gamma}{1 - \gamma} \delta^\alpha \leq M \delta^p$ as $\delta \rightarrow 0$, which is impossible since $p > \alpha > 0$. Therefore, $\alpha = \frac{-\log \gamma}{\log \theta}$ is the largest Holder exponent. By the way, we do not have $V(\delta) = \sum_{t = 0}^\infty \gamma^t |s_t|=\delta \sum_{t = 0}^\infty (\gamma \theta)^t$ since $|s_t| \leq 1$ according to (12).
>
> 3. Regarding the connection between Holder exponents and fractal theory, Proposition C.1 says that when the function $F: \mathbb{R}^N \rightarrow \mathbb{R}$ is $\alpha$-Holder continuous, the Hausdorff dimension of its landscape is bounded above by $\frac{N}{\alpha}$ where $\frac{N}{\alpha}$ is greater than $N$ when $\alpha < 1$. Thus, the landscape can have a fractional dimention between $N$ and $N + 1$, which means that the landscape is truly a fractal.
>
> Another reason why we insist to use the notion of fractal is that when mentioning non-smoothness in optimization, people conventionally think of functions like $f(x) = |x|$ or $f(x) = \max(x, 1)$ that are much smoother than what we have seen in this paper. By associating the landscape with fractals, we encourage researchers to think beyond existing frameworks and explore innovative approaches to tackle this problem, which has barely been discussed before.

---

### Decision · Program_Chairs · 2023-09-21

**Decision:**

Accept (poster)

**Comment:**

**Summary**: This paper shows that for deterministic MDPs with continuous state and action spaces, optimization landscape in the policy space can be extremely non-smooth or fractal. The authors did this by showing that if MDP's maximum Lyapunov exponent at $\theta$ (denoted as $\lambda(\theta)$ ) is larger than $- \log{\gamma} > 1$, then "$J(\theta)$ is not guaranteed (and is very unlikely) to be Lipschitz continuous".

**Strength**:

1. Reviewers all agreed that the paper studied a fundamental problem, and the aim of understanding why and how policy gradient methods often fail in control problems is highly relevant.

2. Characterizing the pathological policy value landscape received little or no attention to date, and the paper provided very interesting ideas and insights, using maximum Lyapunov exponent characterization.

**Weaknesses**:

1. There is a gap between what was proved and the non-smoothness claim.

As mentioned in the paper, "$J(\theta)$ is not guaranteed (and is very unlikely) to be Lipschitz continuous when $\lambda(\theta) > - \log{\gamma}$".

This was mainly criticized by Reviewers Qd1b and dey2. And the direct result is that the prevalence of fractal landscapes in policy optimization is not clear to the reviewers, and nor are the consequences.

From the discussion and the paper, the situation is that there exist examples (Example 4.1) where the largest Holder exponent is $< 1$, which implies that the policy value is non-smooth. However, for the main results (Theorems 4.1 and 4.2), the situation is that the results cannot exclude smooth functions, including but not limited to constant reward functions as mentioned by authors in discussions.

This is indeed a concern, considering the importance of the problem studied and claim made in this paper. The authors also mentioned in the discussion "as long as a reward function can continuously distinguish two separate trajectories ... the objective function will be non-smooth, as shown in experiments", which seems to me they could possibly fill the gap.

The authors should fill the gap, which will make the theoretical results more convincing.

2. Re-writing / re-organizing the paper.

This is also mainly criticized by Reviewers Qd1b and dey2.

The title and main statements both indicate that certain MDPs could have fractal landscapes. However, the discussion of the fractal theory is relegated to the appendix. During rebuttal, the authors also agreed with the reviewers that the writing should be changed. The paper would be largely improved from re-writing / re-organizing.

**Recommendation**:

There are disagreements between reviews after rebuttal. After reading the paper and reviews, as well as discussions, the AC acknowledged that the paper studied an important problem and provided very interesting ideas and insights, while there are indeed two concerns. We recommend the authors to fill the gap in the theory mentioned by reviewers and re-organize the exposition.